# LATENT-INFORMED ENERGY-BASED MODELS WITH COLLABORATIVE GENERATOR TRAINING

## ABSTRACT

Energy-based models (EBMs) have established a distinct niche in generative modeling through their architectural flexibility and expressive density estimation. However, they have yet to achieve mainstream adoption due to their training challenges. In this paper, we propose training latent-informed EBMs that leverage self-supervised representation learning to derive informative target latent variables. This joint space optimization enables the energy function to capture both data distribution and semantic manifold geometry. To avoid long-run MCMC sampling, we introduce an auxiliary generator with specialized training designs for effective energy-generator collaboration. Our training paradigm only requires MCMC sampling in the data space, and the joint energy function learns semantic data–latent relationships directly from real data. Experiments show our approach significantly boosts the generation performance compared to current EBMs with fewer MCMC steps and smaller networks. We also demonstrate the capabilities of our model across multiple tasks, including out-of-distribution detection, conditional sampling, and zero-shot image restoration.

## 1 INTRODUCTION

Generative models have achieved unprecedented rapid development in recent years. Energy-based models (EBMs) (LeCun et al., 2006; Salakhutdinov et al., 2007; Du & Mordatch, 2019), as a class of generative models, occupy a unique position among various generative frameworks due to their huge potential in modeling complex data distributions. With a flexible energy function to directly characterize the underlying probability distribution, EBM can be useful in various tasks such as image and video synthesis (Xie et al., 2019; Zhao et al., 2020), image restoration (Xie et al., 2021a; Gao et al., 2021), compositional generation (Du et al., 2020; 2023), and out-of-distribution (OOD) detection (Yoon et al., 2023; 2021). However, it is notorious for hard training and long-run MCMC sampling (Grathwohl et al., 2021; Nijkamp et al., 2020), leaving a noticeable gap with dominant generative models.

Adversarial EBMs (Geng et al., 2021; 2024) and cooperative learning (Xie et al., 2020) incorporate a generator to speed up sampling and improve generation quality. However, adversarial EBMs are prone to suffering from mode collapse because of their minimax training strategy. Cooperative learning leads to biased generator learning, thereby limiting the potential for learning a robust EBM. Divergence Triangle methods (Han et al., 2019; 2020) extend this co-training scheme to latent-variable models. However, by enforcing exact alignment between the latent representation and the generator's prior distribution, they restrict both generation quality and latent space flexibility, ultimately weakening the energy function. CLEL (Lee et al., 2023) designs a new class of latent-variable EBMs that model the joint distribution using a contrastive latent encoder. This architecture enables the energy function to benefit from the semantically informative latent representation, moving beyond the conventional Gaussian posterior. But its training paradigm is inelegant, and sampling remains slow.

We propose a collaborative training framework that combines latent-informed EBMs (LIEBMs) with auxiliary generator initialization. For each training step, the energy function and generator are updated alternatively. When training LIEBM, the defined energy distribution is optimized in a joint space, where the target latent variables are derived through a pretrained self-supervised latent encoder. This design helps energy function understand the semantic geometry of the data manifold, as shown in Fig.1. Samples from the energy distribution are required for training as negative

samples. We obtain these via generator-predicted initial samples, followed by brief MCMC sampling. An augmentation technique is applied to negative samples to improve the energy function's discrimination of regions that are far away from the data distribution. Our generator learns to approximate the long-run MCMC dynamics through a single-step transformation, thereby enabling efficient short-run sampling that avoids the slow convergence of traditional MCMC approaches. We investigate several designs of generator learning and conduct a thorough comparison among them. Our method improves EBM performance through dedicated collaborative training framework and isolates the generative prior from semantic latent representation, thus avoiding the potential pitfalls of "posterior collapse" (Geng et al., 2023).

Our main contributions are summarized as follows:

1. We introduce a unified and efficient latent-informed EBM, demonstrating the necessity of joint space optimization. Our training paradigm requires MCMC sampling only in the data space, allowing the joint energy function to learn semantic data–latent relationships directly from real samples, rather than between real and synthetic spaces.

2. Our approach utilizes pretrained self-supervised representations as latent variables, maintaining their independence from the generator's prior. This architecture provides semantic guidance to the energy function to tap into full generative potential.

3. We develop a collaborative training framework between the EBM and an auxiliary generator, incorporating key design choices that enable effective collaboration, including a negative-sample augmentation strategy and adaptive generator training schemes.

4. Our method achieves superior sample quality with lightweight architectures, while exhibiting versatile applicability across multiple downstream tasks, including OOD detection, conditional sampling, and zero-shot image restoration.

## 2 PRELIMINARY

Latent-variable EBMs generalize standard EBMs by incorporating a latent variable to model a joint distribution $p_\theta(\mathrm{x}, \mathrm{z})$:

$$p_\theta(\mathrm{x}, \mathrm{z}) = \frac{\exp\left(E_\theta(\mathrm{x}, \mathrm{z})\right)}{Z_\theta}, \quad Z_\theta = \int \exp\left(E_\theta(\mathrm{x}, \mathrm{z})\right) d\mathrm{x} d z, \tag{1}$$

where $Z_\theta$ is the intractable normalizing constant called the partition function. Training latent-variable EBMs primarily relies on maximizing the log-likelihood such that:

$$L(\theta) := \mathbb{E}_{(\mathrm{x},\mathrm{z}) \sim p_{\text{data}}(\mathrm{x},\mathrm{z})} \left[\log p_\theta(\mathrm{x}, \mathrm{z})\right] = \mathbb{E}_{(\mathrm{x},\mathrm{z}) \sim p_{\text{data}}(\mathrm{x},\mathrm{z})} \left[E_\theta(\mathrm{x}, \mathrm{z})\right] - \log Z_\theta. \tag{2}$$

Similar to standard EBMs, the gradient of the training objective can be written as:

$$\frac{\partial L}{\partial \theta} = \mathbb{E}_{(\mathrm{x},\mathrm{z}) \sim p_{\text{data}}(\mathrm{x},\mathrm{z})} \left[\frac{\partial}{\partial \theta} E_\theta(\mathrm{x}, \mathrm{z})\right] - \mathbb{E}_{(\mathrm{x},\mathrm{z}) \sim p_\theta(\mathrm{x},\mathrm{z})} \left[\frac{\partial}{\partial \theta} E_\theta(\mathrm{x}, \mathrm{z})\right]. \tag{3}$$

It requires MCMC sampling from a joint distribution $p_\theta(\mathrm{x}, \mathrm{z})$, which can be challenging in complex high-dimensional space (Xu et al., 2018). Alternatively, Eq.3 can be reformulated to require only sampling from the marginal distribution $p_\theta(\mathrm{x})$:

$$\frac{\partial L}{\partial \theta} = \mathbb{E}_{(\mathrm{x},\mathrm{z}) \sim p_{\text{data}}(\mathrm{x},\mathrm{z})} \left[\frac{\partial}{\partial \theta} E_\theta(\mathrm{x}, \mathrm{z})\right] - \mathbb{E}_{\mathrm{x} \sim p_\theta(\mathrm{x})} \left[\frac{\partial}{\partial \theta} E_\theta(\mathrm{x})\right], \tag{4}$$

where $E_\theta(\mathrm{x}) = \log \int \exp\left(E_\theta(\mathrm{x}, \mathrm{z})\right) dz$ is an available energy function of marginal $p_\theta(\mathrm{x})$. See Appendix A.3 and A.4 for derivation and additional details about EBMs.

## 3 METHOD

Conventional EBMs train the energy function solely in the data space, which poses challenges in high-dimensional settings due to data sparsity and limited distributional information. To address this, we propose a latent-variable EBM with structured latent constraints and generator-assisted

MCMC initialization. The energy function and generator are trained alternatively within each training step. Our formulation needs to solve three fundamental problems: defining a target joint distribution $p_{\text{data}}(\mathrm{x}, \mathrm{z})$ given only observed samples x, constructing a joint energy distribution $p_\theta(\mathrm{x}, \mathrm{z})$ that captures data-latent coupling, and balancing collaborative training between the energy function and generator.

### 3.1 LATENT-INFORMED EBM TRAINING

We define a conditional latent distribution by mapping data samples to latent variables through a latent encoder, i.e., sampling $p_{\text{data}}(\mathrm{z}|\mathrm{x})$ through $h(v(\mathrm{x}))/\|h(v(\mathrm{x}))\|_2, v \sim \mathcal{V}$, where $\mathcal{V}$ is a random operation distribution. Our latent encoder $h$ is pretrained using self-supervised representation learning as a separate stage before EBM training. We observe in Fig.7 that a CLEL-

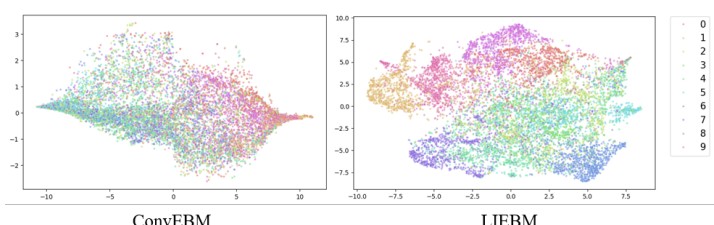

ConvEBM            LIEBM

Figure 1: t-SNE visualization of $f_\phi(\mathrm{x})$ trained on CIFAR-10: conventional EBM with $F(f_\phi(\mathrm{x}))$ as energy function vs. our LIEBM.

style collaborative training approach leads to measurable degradation in the encoder's classification accuracy, which subsequently impairs EBM training. This phenomenon may stem from our generator-initialized EBM samples inadequately covering the true data manifold in the early stage of training, making their latent variables ineffective as negative representations for diversity.

Considering modeling a joint energy distribution, we define our energy function by decomposing the joint density into an implicit data distribution and an explicit latent posterior:

$$E_\theta(\mathrm{x}, \mathrm{z}) = F(f_\phi(\mathrm{x})) + \log p_{\phi,\psi}(\mathrm{z}|\mathrm{x}), \tag{5}$$

$$E_\theta(\mathrm{x}) = \log \int \exp\left(E_\theta(\mathrm{x}, \mathrm{z})\right) d\mathrm{z} = F(f_\phi(\mathrm{x})), \tag{6}$$

where $f_\phi(\mathrm{x})$ is a neural network parameterized by $\phi$, $F$ maps $f_\phi(\mathrm{x})$ to a scalar value, which can be a non-parametric function or a neural network. $p_{\phi,\psi}(\mathrm{z}|\mathrm{x})$ is a probability density parameterized by $(\phi, \psi)$, and $\theta = (\phi, \psi)$. This formulation permits EBM training via Eq.4, requiring only that $p_{\phi,\psi}(\mathrm{z}|\mathrm{x})$ be an explicit density function. Employing Eq.4 as training objective only requires MCMC sampling in data space, avoiding expensive cost in augmented $(\mathrm{x}, \mathrm{z})$ space. Moreover, via Eq.4, our joint energy function learns semantic data-latent relationships directly from real data, which is more reasonable than previous contrastive learning between real and fake spaces in EBMs and GANs.

Our joint energy definition in Eq.5 is a general formulation that theoretically encompasses most existing latent-variable EBM variants. When $p_{\phi,\psi}(\mathrm{z}|\mathrm{x})$ is defined as Gaussian, our $E_\theta(\mathrm{x}, \mathrm{z})$ reduces to conventional formulations (Cui & Han, 2023; Han et al., 2020; Kan et al., 2022); With cosine-similarity form, it's similar to CLEL; Furthermore, with an exponential family form of $E_\theta(\mathrm{x}, \mathrm{z}) = \langle \lambda - f_\phi(\mathrm{x}), \eta(\mathrm{z}) \rangle + B(\lambda)$, our definition reduces to CEBM (Wu et al., 2021). In this case, $E_\theta(\mathrm{x}) = B(\lambda + f_\phi(\mathrm{x})) - B(\lambda)$ and $p_{\phi,\psi}(\mathrm{z}|\mathrm{x}) = \langle \eta(\mathrm{z}), \lambda + f_\phi(\mathrm{x}) \rangle - B(\lambda + f_\phi(\mathrm{x}))$. With $\lambda = (\lambda_1, \lambda_2) = (0, -\frac{1}{2})$, $B(\lambda) = -\frac{\lambda_1^2}{4\lambda_2} - \frac{1}{2}\log(-2\lambda_2)$, and $\eta(z_k) = (z_k, z_k^2)$ for each dimension of z, $p_{\phi,\psi}(\mathrm{z}|\mathrm{x})$ becomes a Gaussian distribution. We empirically compare these choices, and the cosine-similarity is stable and performs significantly better than the others (See Sec.4.5). Therefore, we adopt cosine-similarity form to define posterior $p_{\phi,\psi}(\mathrm{z}|\mathrm{x})$ on a unit sphere:

$$p_{\phi,\psi}(\mathrm{z}|\mathrm{x}) = \frac{\exp\left(\gamma \operatorname{sim}\left(g_\psi\left(f_\phi(\mathrm{x})\right), \mathrm{z}\right)\right)}{Z_\gamma}, \quad \mathrm{z} \sim \mathbb{S}^{d_z - 1} \tag{7}$$

where $\operatorname{sim}(\mathrm{u}, \mathrm{v}) = \mathrm{u}^\top \mathrm{v} / \|\mathrm{u}\|_2 \|\mathrm{v}\|_2$ is the cosine similarity. This definition revisits the conventional use of Gaussian latents with benefits from two critical properties: (1) the normalizing constant $Z_\gamma$ is independent of $\theta$, which can be omitted during training; (2) the scale hyperparameter $\gamma$ controls density magnitudes for training stability.

We optimize our energy function using Eq.4, which requires sampling negative samples from the marginal $p_\theta(\mathrm{x})$. To avoid long MCMC chains, we consider first generating initial samples through a

generator, i.e., $x^0 = G(m), m \sim \mathcal{N}(0, I)$, then refining them with a few MCMC steps from $E_\theta(x^t)$. However, this strategy exhibits a practical limitation: the initial sample distribution progressively becomes closer to the data distribution during training, resulting in the energy function's catastrophic forgetting of low-density regions and earlier discovered modes. To mitigate this problem, we implement a **stochastic augmentation strategy** for negative samples before MCMC sampling. Each negative sample undergoes augmentation with Bernoulli probability $p$, where the augmented transformation $v \sim \mathcal{V}$ follows the same protocol as used for sampling from $p_{\text{data}}(z|x)$. This augmentation technique enables broader exploration of the energy landscape during training, facilitating diversity of MCMC chains. Empirically, this augmentation technique enhances OOD detection for distant outliers with minimal impact on generation quality.

## 3.2 GENERATOR TRAINING

We introduce a generator to initialize MCMC chains via single-step forward propagation. This generator is typically optimized through adversarial training or cooperative learning. We build on cooperative learning as adversarial training would necessitate computationally challenging entropy maximization of the generated distribution $p_g(x)$. Beyond cooperative learning, our framework features a joint energy function and a semantic-aware latent encoder. These architectural advantages allow us to investigate distinct generator training schemes through extensive empirical analysis.

### 3.2.1 ENERGY DISTRIBUTION MATCHING (EM)

Following cooperative learning, the generator can be optimized by minimizing the KL divergence between two joint distributions, $\min \text{KL}\left(p_\theta(x, m) \| p_g(x, m)\right)$, both distributions built from a Gaussian prior $p(m)$ and conditional $p(x|m)$. Under the assumption that $p_g(x|m)$ follows a Gaussian distribution, this objective simplifies to an MSE loss:

$$L_G = \frac{1}{n} \sum_{i=1}^{n} \frac{1}{2\tau^2} \left\| G(m_i) - x_i^T \right\|_2^2, \tag{8}$$

where $i$ denotes $i^{th}$ number of a batch with size $n$. $\tau^2$ is the fixed variance of $p_g(x|m)$. $x_i^T$ is the refined samples by running $T$ steps of MCMC from initial point $x_i^0 = G(m_i)$:

$$x_i^{t+1} = x_i^t + \frac{\delta^2}{2} \nabla_x E_\theta(x_i^t) + \delta \epsilon^t, \quad \epsilon^t \sim \mathcal{N}(0, I) \tag{9}$$

$E_\theta$ is the marginal energy defined in Eq.6. We empirically observe that this marginal energy MCMC performs well across all evaluated datasets.

We also investigate MCMC refinement from the perspective of the joint energy function. We take the basic idea of auxiliary variable MCMC (Brooks et al., 2011; Song & Ou, 2018) to sample in the augmented space $(x, z)$. To circumvent the computational burden of two Markov chains in both data and latent spaces, we employ our latent encoder to perform a single MCMC procedure. Specifically, we first sample initial $x_i^0 = G(m)$, followed by executing MCMC as described below:

$$x_i^{t+1} = x_i^t + \frac{\delta^2}{2} \nabla_x E_\theta(x_i^t, h(x_i^t)) + \delta \epsilon^t, \quad \epsilon^t \sim \mathcal{N}(0, I) \tag{10}$$

This approach is justified because $p_\theta(z|x)$ is learned to match $p_{\text{data}}(z|x)$ during EBM training, while constraining x within the latent space reduces the search space and improves efficiency. We observe that this joint energy refinement accelerates training in the early stage, but ultimately underperforms marginal MCMC when dealing with multimodal distributions.

### 3.2.2 ENERGY AND REAL DISTRIBUTION MATCHING (ERM)

Dual-MCMC (Cui & Han, 2023) highlights that energy distribution matching may induce biased generator learning because it solely aligns with the energy distribution without direct access to training data. Inspired by Dual-MCMC, also leveraging our latent encoder, we optimize the generator to match both the energy and real data distribution, yielding a more informative initialization.

$$L_G = \omega_1 \text{KL}(p_\theta(x, m) \| p_g(x, m)) + \omega_2 \text{KL}(p_{\text{data}}(x, z) \| p_g(x, z)), \tag{11}$$

where $\omega_1$ and $\omega_2$ denote the importance weighting between two divergence components. The first term is equal to Eq.8. For the second term, we define $p_g(x, z)$ as:

$$p_g(x, z) = \int p(m) p_g(x, z|m) dm, \tag{12}$$

$$\log p_g(x, z|m) = \log p_g(x|m) + \rho \sin(z, h(G(m))) - \log Z_\rho. \tag{13}$$

We define $p_g(x, z)$ in this way to facilitate both latent alignment and pixel-level fidelity. The second term in Eq.11 can be optimized by the classic evidence lower bound (ELBO):

$$-\mathbb{E}_{p_{\text{data}}(x,z)} \mathbb{E}_{q_\alpha(m|x)} \left[ \log p_g(x, z|m) - \log \frac{q_\alpha(m|x)}{p(m)} \right] \tag{14}$$

where $q_\alpha$ denotes an inference model parameterized by $\alpha$ and jointly trained with the generator. This method introduces an extra network, while increasing training complexity, this autoencoder-based architecture would be necessary for applications such as image restoration.

Table 1: Generative performance on CIFAR-10. "w/o MCMC" denotes direct sampling from the generator without energy-based refinement via MCMC sampling.

| Model | NFE↓ | FID↓ | IS↑ | Model | NFE↓ | FID↓ | IS↑ |
|---|---|---|---|---|---|---|---|
| **Likelihood-based** | | | | **EBM-based** | | | |
| PixelCNN (Oord et al., 2016) | 1024 | 65.9 | 4.60 | IGEBM (Du & Mordatch, 2019) | 60 | 38.2 | 6.78 |
| Glow (Kingma & Dhariwal, 2018) | 1 | 48.9 | 3.92 | joint Triangle (Han et al., 2020) | 1 | 30.10 | 7.17 |
| VAE (Kingma & Welling, 2014) | 1 | 115.8 | 3.8 | CoopNets (Xie et al., 2020) | 51 | 33.61 | 6.55 |
| NVAE (Vahdat & Kautz, 2020) | 1 | 51.67 | 5.51 | EBMBB (Geng et al., 2021) | 1 | 28.63 | 7.45 |
| **GAN-based** | | | | VAEBM (Xiao et al., 2021) | 16 | 12.19 | 8.43 |
| | | | | DRL (Gao et al., 2021) | 180 | 9.58 | 8.30 |
| SN-GAN (Miyato et al., 2018) | 1 | 21.7 | 8.22 | CoopFlow (Xie et al., 2022) | 31 | 15.80 | – |
| BigGAN (Brock et al., 2019) | 1 | 14.73 | 9.22 | Hat EBM (Hill et al., 2022) | 51 | 19.30 | – |
| StyleGAN2 w/ ADA(Karras et al., 2020) | 1 | 2.92 | 9.83 | CLEL-Large (Lee et al., 2023) | 1200 | 8.61 | – |
| DDGAN (Xiao et al., 2022) | 4 | 3.75 | 9.63 | Dual-MCMC (Cui & Han, 2023) | 31 | 9.26 | 8.55 |
| ACT (Kong et al., 2024) | 1 | 6.0 | 9.15 | DDAEBM (Geng et al., 2024) | 4 | 4.82 | 8.86 |
| **Diffusion-based** | | | | CDRL (Zhu et al., 2024) | 96 | 4.31 | – |
| | | | | EC-VAE (Luo et al., 2024) | 1 | 5.20 | – |
| NCSN-v2 (Song & Ermon, 2020) | 1000 | 10.87 | 8.40 | **Ours** | | | |
| DDPM (Ho et al., 2020) | 1000 | 3.17 | 9.46 | LIEBM-EM w/o MCMC | 1 | 4.96 | 9.82 |
| NCSN++ (Song et al., 2021) | 2000 | 2.20 | **9.89** | LIEBM-EM | 16 | **4.26** | **10.02** |
| EDM (Karras et al., 2022) | 35 | **2.04** | 9.84 | LIEBM-ERM w/o MCMC | 1 | 6.16 | 9.41 |
| Flow Matching (Lipman et al., 2023) | 142 | 6.35 | – | LIEBM-ERM | 16 | 4.96 | 9.64 |
| Consistency Models (Song et al., 2023) | 1 | 8.70 | 8.49 | | | | |

## 4 EXPERIMENTS

We conduct comprehensive experiments to evaluate our proposed method under various scenarios, including unconditional image generation, OOD detection, conditional sampling, and zero-shot image restoration. We consider two options for $\mathcal{V}$ for sampling $p_{\text{data}}(z|x)$: (i) the standard random augmentations commonly used in self-supervised representation learning, and (ii) adding minor uniform noise via $x = \frac{255}{256}x + z$, where $z \sim \mathcal{U}(0, \frac{1}{256})$. Both choices yield similar performance, and we adopt the first one to introduce more stochasticity into the latent variables. For our pretrained latent encoder, we evaluate three normalized self-supervised representation learning methods: SimCLR (Chen et al., 2020), BYOL (Grill et al., 2020), and W-MSE (Ermolov et al., 2021). We select SimCLR[1] since it achieves the best generation performance on CIFAR-10. We adopt the same architecture as Dual-MCMC for our generator and inference model. We use the energy function backbone from Dual-MCMC as our $f_\phi$ in $E_\theta(x, z)$, while our $g_\psi$ in $p_{\phi,\psi}(z|x)$ follows the projection head architecture of the latent encoder, with Batch Normalization (Ioffe & Szegedy, 2015) removed. We apply Exponential Moving Average (EMA) with a decay rate of 0.9999 to improve generation

---

[1]We implement SimCLR using the official code of W-MSE: https://github.com/htdt/self-supervised

quality. We denote our model with EM generator training as LIEBM-EM, and ERM as LIEBM-ERM. For the EM setting, MCMC with marginal energy (Eq.9) outperforms joint energy matching (Eq.10), so we use Eq.9 for LIEBM-EM.

Table 2: Generative performance on CelebA 64

| Model | FID ↓ |
|---|---|
| SN-GAN (Miyato et al., 2018) | 6.1 |
| COCO-GAN (Lin et al., 2019) | 4.0 |
| NVAE (Vahdat & Kautz, 2020) | 14.74 |
| NCSNv2 (Song & Ermon, 2020) | 26.86 |
| DDPM (Ho et al., 2020) | 3.93 |
| **EBM-based** | |
| DRL (Gao et al., 2021) | 5.98 |
| VAEBM (Xiao et al., 2021) | 5.31 |
| CLEL (Lee et al., 2023) | 8.34 |
| Dual MCMC (Cui & Han, 2023) | 5.15 |
| EC-VAE (Luo et al., 2024) | **2.71** |
| LIEBM-EM | 3.44 |
| LIEBM-ERM | 2.97 |

Table 3: Generative performance on CelebA-HQ 256.

| Model | FID ↓ |
|---|---|
| GLOW (Kingma & Dhariwal, 2018) | 68.93 |
| NVAE (Vahdat & Kautz, 2020) | 45.11 |
| VQGAN (Esser et al., 2021) | 10.2 |
| DDGAN (Xiao et al., 2022) | 7.64 |
| Score SDE (Song et al., 2021) | **7.23** |
| **EBM-based** | |
| VAEBM (Xiao et al., 2021) | 20.38 |
| Dual MCMC (Cui & Han, 2023) | 15.89 |
| CDRL (Zhu et al., 2024) | 10.74 |
| EC-VAE (Luo et al., 2024) | 12.35 |
| LIEBM-EM | 10.08 |
| LIEBM-ERM | 8.76 |

Table 4: Generative performance on ImageNet 32.

| Model | FID ↓ |
|---|---|
| PixelCNN (Oord et al., 2016) | 40.51 |
| DDPM++ (Kim et al., 2021) | 8.42 |
| Flow Matching (Lipman et al., 2023) | 5.02 |
| **EBM-based** | |
| CF-EBM (Zhao et al., 2020) | 26.31 |
| EBM-CD (Du et al., 2021) | 32.48 |
| CLEL-Large (Lee et al., 2023) | 15.47 |
| CDRL (Zhu et al., 2024) | 9.35 |
| EC-VAE (Luo et al., 2024) | 5.76 |
| $EBM_{MI+diff}$ (Geng et al., 2025) | 6.57 |
| LIEBM-EM | **4.54** |
| LIEBM-ERM | 4.98 |

## 4.1 UNCONDITIONAL IMAGE GENERATION

We showcase our model's capabilities in unconditional image generation on standard datasets involving CIFAR-10 (Krizhevsky et al., 2009), ImageNet 32 (Deng et al., 2009), CelebA 64 (Liu et al., 2015b), and CelebA-HQ 256 (Liu et al., 2015a). For quantitative results, we adopt the commonly used Fréchet inception distance (FID) and Inception Score (IS) to evaluate sample fidelity and the number of function evaluations (NFE) to evaluate sampling efficiency. We show qualitative results in Fig.3 and quantitative results in Tabs.1-4 [2]. Fig.2 shows FID vs. network scale on CIFAR-10. Fig.5 further illustrates our method's sampling efficiency by comparing inference time and FID across different generative models.

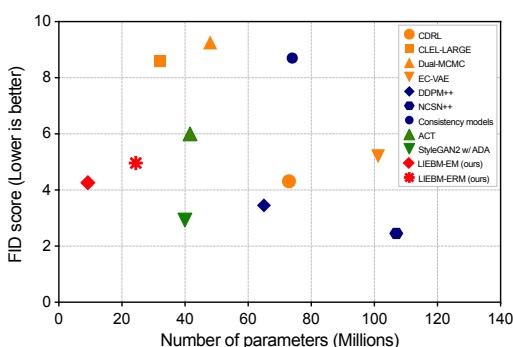

Figure 2: Param count vs. FID on CIFAR-10.

Our model achieves optimal results on most datasets, with near-optimal performance on CelebA 64 among EBMs. On CIFAR-10, our LIEBM-EM outperforms state-of-the-art CDRL, despite CDRL having 5× the number of parameters and requiring 6× MCMC steps.

Our method achieves significant improvements over CLEL with much faster sampling, demonstrating the effectiveness of our carefully designed collaborative training between the energy function and generator. Our model also outperforms Dual-MCMC and EC-VAE by a large margin on most datasets with fewer network parameters and MCMC steps, validating that our latent-informed scheme can further improve generation. Notably, for single-step generation, our model surpasses strong diffusion baselines, including Consistency Model and its adversarial variant ACT. Moreover, our model achieves competitive performance with advanced GANs and Diffusion Models while using 5-10× fewer parameters. Our model gets the best IS score on CIFAR-10 and is the first EBM to beat Flow Matching on ImageNet 32.

---

[2]Since baselines for ImageNet 32, CelebA 64, and CelebA-HQ 256 are less established than CIFAR-10, we compare using FID and commonly reported baselines.

Table 5: Inference time vs. FID on CIFAR-10.

| Method | Time(s)↓ | FID↓ | GPU-Type |
|---|---|---|---|
| NVAE | 0.36 | 50.97 | V100 |
| StyleGAN2 w/ ADA | 0.04 | 2.92 | V100 |
| DDPM | 80.5 | 3.17 | V100 |
| NCSN++ | 423.2 | 2.20 | V100 |
| EBM-Based | | | |
| VAEBM | 8.79 | 12.19 | V100 |
| EC-VAE | 0.21 | 5.20 | RTX 2080 Ti |
| CLEL | 82.05 | 8.61 | RTX 2080 Ti |
| Dual MCMC | 9.32 | 9.26 | RTX 2080 Ti |
| LIEBM w/o MCMC(ours) | 0.08 | 4.96 | RTX 2080 Ti |
| LIEBM(ours) | 1.24 | 4.26 | RTX 2080 Ti |

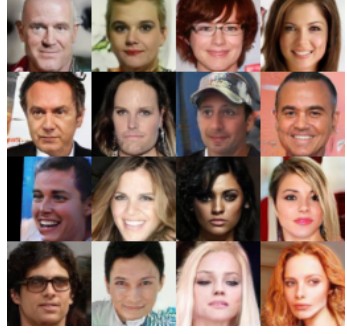

(a) CIFAR-10        (b) ImageNet 32        (c) CelebA 64

(d) CelebA-HQ 256

Figure 3: Samples generated by LIEBM with MCMC refinement. Select models based on FID: LIEBM-EM for CIFAR-10/ImageNet-32; LIEBM-ERM for CelebA-64/CelebA-HQ-256.

## 4.2 OUT-OF-DISTRIBUTION DETECTION

We evaluate our model's density modeling through OOD detection on CIFAR-10 and ImageNet 32, using their unseen test sets as inliers and other datasets as outliers. We use the standard AUROC metric with a joint energy score inspired by CLEL:

$$s(\mathbf{x}) := F\left(f_\theta(\mathbf{x})\right) + \gamma \sim \left(g_\psi\left(f_\phi(\mathbf{x})\right), h(\mathbf{x})\right). \tag{15}$$

Results are shown in Tabs.6 and 7. We observe that the joint energy score improves OOD detection on most datasets (with only slight degradation on SVHN), demonstrating enhanced robustness to diverse OOD samples through joint space modeling. On CIFAR-10, our model consistently performs at the top tier among EBMs and matches specialized OOD methods. Notably, our model shows significant improvement on CIFAR-100, which is challenging due to the similarity between CIFAR-100 and CIFAR-10. We reproduce Dual-MCMC and hat-EBM using their energy outputs as AUROC decision values. Our model exhibits strong performance on the challenging SVHN and Constant datasets for ImageNet 32, where likelihood-based methods such as VAE, GLOW, and PixelCNN typically fail at outlier detection.

Table 6: AUROC with CIFAR-10 as in-distribution.

| Method | SVHN | Constant | CIFAR-100 | CelebA |
|---|---|---|---|---|
| PixelCNN++ (Salimans et al., 2017) | 0.32 | 0.71 | 0.63 | – |
| GLOW (Kingma & Dhariwal, 2018) | 0.24 | – | 0.55 | 0.57 |
| NVAE (Vahdat & Kautz, 2020) | 0.44 | 0.65 | 0.49 | 0.68 |
| JEM (Duvenaud et al., 2020) | 0.67 | – | 0.67 | 0.75 |
| DRL (Gao et al., 2021) | 0.88 | 0.99 | 0.44 | 0.64 |
| hatEBM (Hill et al., 2022) | 0.75 | 0.36 | 0.63 | 0.62 |
| CLEL (Lee et al., 2023) | 0.98 | – | 0.72 | 0.77 |
| Dual-MCMC(Cui & Han, 2023) | 0.62 | 0.32 | 0.54 | 0.59 |
| **Specialized OOD methods** | | | | |
| OOD EBM (Liu et al., 2020) | 0.91 | – | **0.87** | **0.78** |
| MPDR-S (Yoon et al., 2023) | **0.99** | **0.9996** | 0.56 | 0.73 |
| LIEBM-EM $\left(f_\theta(\mathrm{x})\right)$ | 0.96 | 0.67 | 0.66 | 0.68 |
| LIEBM-EM | 0.95 | 0.97 | 0.82 | 0.77 |
| LIEBM-ERM $\left(f_\theta(\mathrm{x})\right)$ | 0.94 | 0.76 | 0.68 | 0.58 |
| LIEBM-ERM | 0.95 | 0.96 | 0.82 | 0.75 |

Table 7: AUROC with ImageNet 32 as in-distribution. $\left(f_\theta(\mathrm{x})\right)$ means $f_\theta(\mathrm{x})$ serves as the decision function.

| Method | SVHN | Constant | FMNIST | CelebA |
|---|---|---|---|---|
| DAE (Vincent et al., 2008) | 0.10 | 0.07 | 0.991 | 0.43 |
| VAE (Kingma & Welling, 2014) | 0.13 | 0.03 | 0.95 | 0.55 |
| WAE (Tolstikhin et al., 2018) | 0.08 | 0.07 | 0.991 | 0.36 |
| PixelCNN++ (Salimans et al., 2017) | 0.03 | 0.00 | 0.004 | 0.24 |
| GLOW (Kingma & Dhariwal, 2018) | 0.17 | 0.41 | 0.86 | 0.48 |
| CLEL (Lee et al., 2023) | 0.96 | 0.83 | 0.54 | 0.74 |
| **Specialized OOD methods** | | | | |
| NAE (Yoon et al., 2021) | 0.985 | 0.97 | **0.994** | **0.95** |
| LIEBM-EM $\left(f_\theta(\mathrm{x})\right)$ | **0.99** | 0.97 | 0.40 | 0.48 |
| LIEBM-EM | 0.984 | **0.99** | 0.896 | 0.52 |
| LIEBM-ERM $\left(f_\theta(\mathrm{x})\right)$ | **0.99** | 0.93 | 0.35 | 0.45 |
| LIEBM-ERM | 0.985 | **0.99** | 0.868 | 0.54 |

### 4.3 CONDITIONAL SAMPLING

We also investigate conditional sampling with our latent representation as labels. Unlike CLEL, we employ a generator as an initializer, which offers faster sampling but requires the generator to produce high-quality initial samples. Therefore, similar to the ERM setting, we train an inference model to form an autoencoder with the generator under our EM framework, enabling us to obtain reconstructions from the input for initialization. We train our inference model using a variant of ELBO loss in the latent space to ensure detailed clarity and sharpness while preserving semantic similarity (See Appendix A.6 for more results). Specifically, we use Eq.14 to train our inference model, but omitting the pixel-level reconstruction term $\log p_g(\mathrm{x}|\mathrm{m})$. We split our generation into two components $G(\mathrm{m}) + Y$. Following CLEL, we obtain the class representation $\overline{z_\mathrm{c}}$ for each class $c$, defined as the normalized average of latent representation across all images in class $c$. We draw an initialization $\overline{\mathrm{x_c}}$ as initial $G(\mathrm{m})$ by averaging all augmented images from each class. Then we iteratively optimize $Y$ and m by performing MCMC sampling from $E_\theta(G(\mathrm{m}) + Y, \overline{z_\mathrm{c}})$ and using the inference model conditioned on $G(\mathrm{m}) + Y$, respectively. From Fig.4 we can see that the EM setting is able to generate diverse samples with clear details for each class, whereas the ERM setting, while capable of generating some feature elements of the given class, fails to produce identifiable subjects. This is caused by the ELBO component in ERM training, which provides pixel-level reconstruction but produces blurry, low-sharpness results.

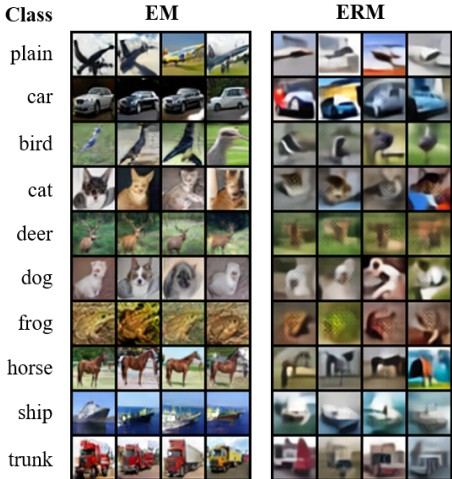

Figure 4: Conditional generated sample on CIFAR-10.



Figure 5: Qualitative results of zero-shot image restoration on CelebA-HQ 256.

## 4.4 IMAGE RESTORATION

We also present the application of our method in zero-shot image restoration tasks, including colorization and $8\times$ super-resolution. We conduct experiments on CelebA-HQ 256 with ERM setting, since we need pixel-level restoration. Following Luo et al. (2024); Wang et al. (2023), we also use a linear operator $A$ to get the degraded image $y = \mathbf{A}x$ and utilize its pseudo-inverse $\mathbf{A}^\dagger$ to derive the initial estimate $\hat{\mathrm{x}} = \mathbf{A}^\dagger y$. We obtain initial $\mathrm{m}^0$ using our inference model with input $\hat{\mathrm{x}}$. Inspired by Luo et al. (2024), we use the following joint function to refine $\mathrm{m}$ using MCMC sampling:

Table 8: Quantitative results of zero-shot image restoration on CelebA-HQ 256.

| **Model** | Colorization PSNR↑ / SSIM↑ | $8\times$ SR PSNR↑ / SSIM↑ |
|---|---|---|
| $G(\mathrm{m}^0)$ | 20.64 / 0.66 | 22.62 / 0.67 |
| $G(\mathrm{m})$ | 22.02 / 0.70 | 24.16 / 0.70 |
| $G(\mathrm{m}) + Y$ | 25.25 / 0.94 | 24.55 / 0.71 |

$$p_{g,\theta}(\mathbf{A}^\dagger y, \mathrm{m}) \propto \exp\left(E_\theta\left(G(\mathrm{m}), h\left(G(\mathrm{m})\right)\right)\right) p(\mathrm{m}) p\left(\mathbf{A}^\dagger y \mid \mathbf{A}^\dagger \mathbf{A} G(\mathrm{m})\right) \tag{16}$$

After refining $\mathrm{m}$, we update $Y$ using MCMC sampling with $G(\mathrm{m})$ fixed:

$$p_{g,\theta}(\mathbf{A}^\dagger y, Y) \propto \exp\left(E_\theta\left(G(\mathrm{m}) + Y, h\left(G(\mathrm{m}) + Y\right)\right)\right) p\left(\mathbf{A}^\dagger y \mid \mathbf{A}^\dagger \mathbf{A}\left(G(\mathrm{m}) + Y\right)\right) \tag{17}$$

We employ $\tilde{\mathrm{x}} = G(\mathrm{m}) + Y$ as our restoration solution. The qualitative results are shown in Fig.5 and the corresponding PSNR and SSIM metrics are reported in Tab.8. We can observe that with the help of joint energy distribution, our model can successfully restore those images with high quality and consistency after refinement on $\mathrm{m}$ and $Y$.

## 4.5 ABLATION STUDY

Fig.6 tracks FID scores during training for the ablation study on CIFAR-10. Tab.9 shows their corresponding OOD performance. It can be seen that traditional EBM training without latent variables can not converge, no matter how the generator training is designed. Training with a pretrained latent encoder improves both generation performance and OOD robustness while yielding a better latent encoder with enhanced semantic separability, as shown in Fig.7. Augmentation technique can improve OOD results on Constant Dataset with negligible generation degradation. Generator training with Eq.10 for MCMC refinement (EJM) can get better results for the first stage, but finally slightly worse than EM setting (Eq.9). In particular, EJM tends to collapse towards the end of training on ImageNet 32. Hence, we recommend employing the EM setting. Combining Tabs.1-4, we can see that our EM setting achieves better performance than ERM on multi-class datasets such as CIFAR-10 and ImageNet, while for few-modal datasets like CelebA and CelebA-HQ, ERM performs better.

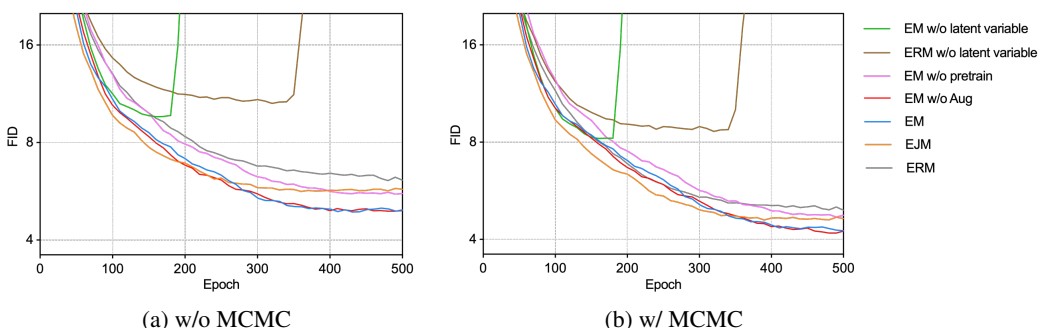

(a) w/o MCMC      (b) w/ MCMC

Figure 6: FID with different settings on CIFAR-10. "w/o MCMC" means direct sampling from the generator without MCMC refinement.

**Generality with different $\mathcal{V}$ choices and energy forms** Fig.8 compares different $\mathcal{V}$ choices and energy forms. For $\mathcal{V}$ choices, we observe similar performance regardless of using random augmentations or uniform noise. This shows that our method is agnostic to the design of $\mathcal{V}$. For energy forms, while the Gaussian distribution is the conventional choice for modeling explicit posterior

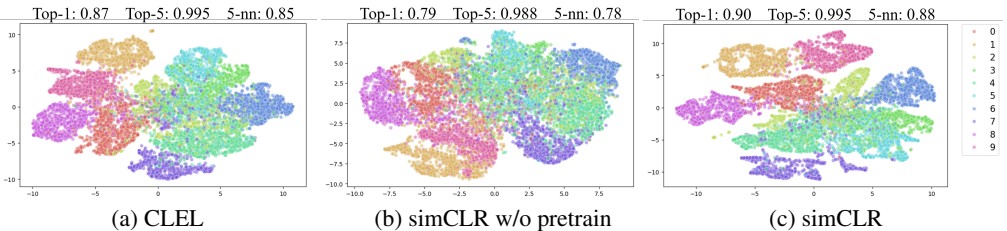

(a) CLEL      (b) simCLR w/o pretrain      (c) simCLR

Figure 7: t-SNE visualization of latent representation on CIFAR-10 test set. Accuracies of a linear classifier (Top-1 & Top-5) and a 5-nearest neighbors classifier are shown above each subfigure.

Table 9: AUROC under different settings with CIFAR-10 as in-distribution.

| Method | SVHN | Constant | CIFAR-100 | CelebA |
|---|---|---|---|---|
| EM w/o pretrain | **0.97** | **0.997** | 0.75 | 0.71 |
| EM w/o Aug | 0.95 | 0.88 | **0.82** | 0.76 |
| EM | 0.95 | 0.97 | **0.82** | **0.77** |
| EJM | 0.95 | 0.94 | 0.81 | 0.73 |
| ERM | 0.95 | 0.96 | **0.82** | 0.75 |

Table 10: Performance with different normalized SSRL methods.

| Method | FID | AUROC | | | |
|---|---|---|---|---|---|
| | | SVHN | Constant | FMNIST | CelebA |
| BYOL | 5.23 | **0.96** | 0.98 | **0.85** | **0.81** |
| W-MSE | 5.16 | 0.93 | **0.99** | 0.83 | 0.77 |
| SimCLR | **4.26** | 0.95 | 0.97 | 0.82 | 0.77 |

$p_{\phi,\psi}(z|x)$ (Han et al., 2019; Kan et al., 2022), we observe its sensitive variance effects. Thus, we constrain the normalized mean and log-variance to $[-1, 1]$ for the Gaussian posterior, and fix the variance for CEBM. Both perform better than the Gaussian-posterior Dual MCMC, yet remain inferior to the cosine-similarity posterior. Moreover, by fixing the variance, the squared Euclidean distance $\|z_1 - z_2\|_2^2 = 2 - 2\sim(z_1, z_2)$, cosine-similarity form naturally induces a spherical Gaussian on the unit sphere. This shows that using a cosine-similarity posterior and decoupling the implicit data energy from the explicit posterior, as in Eq.5, offers greater flexibility and easier control.

**Adaptability to various self-supervised representation learning methods.** Our framework theoretically can be applied to any normalized self-supervised representation learning (SSRL) method. To verify our model's adaptability, we choose two other classic normalized SSRL methods, BYOL and W-MSE, to pretrain our latent encoder. Tab.10 reports FID and AUROC metrics for different SSRL methods, confirming that our LIEBM scales well to various SSRL methods.

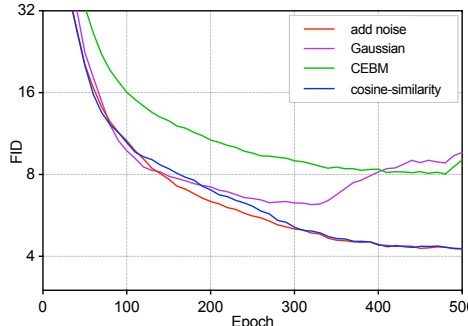

Figure 8: FID with different $\mathcal{V}$ choices and energy forms on CIFAR-10. "add noise" means adding uniform noise to construct $\mathcal{V}$. "Gaussian" and "cosine-similarity" mean two posterior choices.

## 5 CONCLUSION

In this paper, we propose LIEBM, a collaborative training scheme that jointly learns a latent-informed EBM and its generator initializer. We leverage pretrained self-supervised representations as our target latent variables to guide the energy function in capturing the semantic structure of the data manifold. Our model narrows the gap between EBMs and mainstream generative models while retaining the benefits of lightweight architectures. It also excels in various downstream tasks, such as OOD detection, conditional sampling, and zero-shot image restoration. Additionally, our framework could be extended to multi-modal large models by treating the joint space as a multi-modal space and replacing SSRL methods with advanced modal-alignment techniques such as CLIP and ALBEF. We hope our work brings to light the profound potential of EBMs as mainstream generative models and stimulate active research in this area.

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

## A  APPENDIX

### A.1  LLM USAGE

We employ large language models (LLMs) to assist with language polishing and grammar improvement throughout the paper. For the Related Work section, we leverage LLMs to help synthesize brief summaries of related research publications. We also use LLMs to help generate some simple experiment code and LaTeX formatting code for figures and tables. We have verified and validated all contents made by LLMs and take full responsibility for our submission.

### A.2 RELATED WORK

Energy-based models (EBMs) represent a powerful class of generative models that offer explicit unnormalized density estimation and architectural flexibility. Traditional EBM training relies on maximum likelihood estimation (MLE) with Markov Chain Monte Carlo (MCMC) sampling, particularly Langevin dynamics. However, noise-initialized Langevin dynamics often suffer from slow convergence and computational inefficiency (Song & Kingma, 2021). Several techniques have been proposed to alleviate the expensive MCMC, such as Persistent Contrastive Divergence (PCD) (Tieleman, 2008), adding a replay buffer (Du & Mordatch, 2019), short-run MCMC (Nijkamp et al., 2019), et.al. Nevertheless, these approaches remain inefficient as they still require hundreds to thousands of MCMC steps. Cooperative learning methods (Xie et al., 2020; 2021b; 2022; Hill et al., 2022) introduce a generator as a fast initializer learned to amortize long-run MCMC. Adversarial EBMs (Kumar et al., 2019; Geng et al., 2021; Grathwohl et al., 2021; Wan et al., 2025) form a minimax game between the energy function and the introduced generator to enable MCMC-free training. Some advances link connections between EBMs and other generative models to benefit from their strengths, such as VAE (Xiao et al., 2021; Luo et al., 2024), flow-based models (Nijkamp et al., 2022; Gao et al., 2020), and diffusion-based models (Gao et al., 2021; Zhu et al., 2024; Geng et al., 2024). Some recent works (Neklyudov et al., 2023; Loo et al., 2025; Balcerak et al., 2025) formulate EBMs from a vector field perspective, but slow sampling remains an issue.

Latent-variable EBMs define an energy function to characterize the joint density over data and latent variables. CLEL (Lee et al., 2023) leverages contrastive representation learning to learn meaningful latent structures that subsequently guide the EBM training. CEBM (Wu et al., 2021) decomposes the joint density into an intractable data distribution and a tractable latent posterior, providing VAE-like functionality while preserving EBM interpretability and density estimation. Divergence Triangle (Han et al., 2019; 2020) and Dual-MCMC (Cui & Han, 2023) build a unified framework that employs divergence triangle formulations to seamlessly integrate energy function, generator, and inference model through minimizing KL divergences between joint distributions. We focus on collaborative learning between the generator and latent-variable EBM, decoupling the latent distribution from the generator's prior to retain informative latent representations.

### A.3 PRELIMINARY OF EBMS

Let $\mathcal{X}$ be the data space and $p_{\text{data}}(\text{x})$ be true data distribution. An EBM defines a probability distribution through an energy function $E_\theta : \mathcal{X} \to \mathbb{R}$ parameterized by $\theta$,

$$p_\theta(\text{x}) = \frac{\exp\left(E_\theta(\text{x})\right)}{Z_\theta}, \quad Z_\theta = \int \exp\left(E_\theta(\text{x})\right) d\text{x}, \tag{18}$$

where $Z_\theta$ is the intractable normalizing constant. EBMs primarily rely on maximizing the log-likelihood for training such that:

$$L(\theta) := \mathbb{E}_{\text{x} \sim p_{\text{data}}(\text{x})}\left[\log p_\theta(\text{x})\right] = \mathbb{E}_{\text{x} \sim p_{\text{data}}(\text{x})}\left[E_\theta(\text{x})\right] - \log Z_\theta. \tag{19}$$

The gradient of $L(\theta)$ can be derived as:

$$\frac{\partial L}{\partial \theta} = \mathbb{E}_{\text{x} \sim p_{\text{data}}(\text{x})}\left[\frac{\partial}{\partial \theta}E_\theta(\text{x})\right] - \mathbb{E}_{\text{x} \sim p_\theta(\text{x})}\left[\frac{\partial}{\partial \theta}E_\theta(\text{x})\right]. \tag{20}$$

Eq.20 requires MCMC sampling from energy distribution $p_\theta(\text{x})$, which can be achieved by Langevin dynamics (Welling & Teh, 2011):

$$\text{x}^{t+1} = \text{x}^t + \frac{\delta^2}{2}\nabla_{\text{x}}E_\theta(\text{x}^t) + \delta\epsilon^t, \tag{21}$$

where $t$ indexes the time step, $\delta$ is the step size, and $\epsilon \sim \mathcal{N}(0, I)$. For small enough $\epsilon$ and large enough $t$, the distribution of $\text{x}^t$ weakly converges to the energy distribution $p_\theta(\text{x})$ regardless of the initial distribution of $\text{x}^0$ (Raginsky et al., 2017; Xu et al., 2018; Neal et al., 2011).

### A.4 DERIVATION OF EQ.4

From Eq.3, we have

$$\frac{\partial L}{\partial \theta} = \mathbb{E}_{(\text{x,z}) \sim p_{\text{data}}(\text{x,z})}\left[\frac{\partial}{\partial \theta}E_\theta(\text{x}, \text{z})\right] - \mathbb{E}_{(\text{x,z}) \sim p_\theta(\text{x,z})}\left[\frac{\partial}{\partial \theta}E_\theta(\text{x}, \text{z})\right] \tag{22}$$

Since $E_\theta(\mathrm{x}) = \log \int \exp\left(E_\theta(\mathrm{x}, z)\right) dz$, then $p_\theta(\mathrm{x}) = \int p_\theta(\mathrm{x}, z) dz = \frac{\exp(E_\theta(\mathrm{x}))}{Z_\theta}$, thus $E_\theta(\mathrm{x})$ is an available energy function of marginal $p_\theta(\mathrm{x})$. We can obtain:

$$p_\theta(\mathrm{x}, \mathrm{z}) = \frac{\exp\left(E_\theta(\mathrm{x}, \mathrm{z})\right)}{Z_\theta} = \frac{\exp(E_\theta(\mathrm{x}))}{Z_\theta} p_\theta(\mathrm{z}|\mathrm{x})$$

$$E_\theta(\mathrm{x}, \mathrm{z}) = E_\theta(\mathrm{x}) + \log p_\theta(\mathrm{z}|\mathrm{x}) \tag{23}$$

Substituting Eq.23 into the second term of Eq.22 yields:

$$\mathbb{E}_{(\mathrm{x},\mathrm{z})\sim p_\theta(\mathrm{x},\mathrm{z})}\left[\frac{\partial}{\partial\theta}E_\theta(\mathrm{x}, \mathrm{z})\right] = \mathbb{E}_{\mathrm{x}\sim p_\theta(\mathrm{x})}\left[\frac{\partial}{\partial\theta}E_\theta(\mathrm{x})\right] + \mathbb{E}_{\mathrm{x}\sim p_\theta(\mathrm{x},\mathrm{z})}\left[\frac{\partial}{\partial\theta}\log p_\theta(\mathrm{z}|\mathrm{x})\right]$$

$$= \mathbb{E}_{\mathrm{x}\sim p_\theta(\mathrm{x})}\left[\frac{\partial}{\partial\theta}E_\theta(\mathrm{x})\right] \tag{24}$$

The second equality follows from:

$$\mathbb{E}_{\mathrm{x}\sim p_\theta(\mathrm{x},\mathrm{z})}\left[\frac{\partial}{\partial\theta}\log p_\theta(\mathrm{z}|\mathrm{x})\right] = \mathbb{E}_{\mathrm{x}\sim p_\theta(\mathrm{x})}\left[\int \frac{\partial}{\partial\theta}p_\theta(\mathrm{z}|\mathrm{x})dz\right] = 0 \tag{25}$$

Plugging Eq.24 in Eq.22, we can get Eq.4.

## A.5 TRAINING PROCEDURE OF LIEBM

---
**Algorithm 1** LIEBM training

---
**Require:** a latent-variable EBM $E_\theta(\mathrm{x}, \mathrm{z})$, a Generator $G$, an inference model $q_\alpha$, a pretrained latent encoder $h$, an augmentation distribution $\mathcal{V}$, hyperparameters $\tau, p$.

1: **for** # training iterations **do**
2:     Sample $\{\mathrm{x}_i\}_{i=1}^n \sim p_{\mathrm{data}}(\mathbf{x})$.
3:     Sample $\mathrm{z}_i = h(v(\mathrm{x}_i))/\|h(v(\mathrm{x}_i))\|_2, v \sim \mathcal{V}$ to get sample pairs $\{\mathrm{x}_i, \mathrm{z}_i\}_{i=1}^n \sim p_{\mathrm{data}}(\mathrm{x}, \mathrm{z})$
    ▷ Generator training
4:     Sample generated examples $\{G(\mathrm{m}_i)\}_{i=1}^n$, where $\mathrm{m}_i \sim \mathcal{N}(0, I)$
5:     Obtain refined samples $\{\tilde{\mathrm{x}}_i^T\}_{i=1}^n \sim p_\theta(\mathbf{x})$ initialized from $\tilde{\mathrm{x}}_i^0 = G(\mathrm{m}_i)$ using Eq.9 or Eq.10.
6:     Compute $L_G = \frac{1}{n}\sum_{i=1}^n \frac{1}{2\tau^2}\left\|G(\mathrm{m}_i) - \tilde{\mathrm{x}}_i^T\right\|_2^2$
7:     **if** using ERM **then**
8:         Sample $\tilde{\mathrm{m}}_i \sim q_\alpha(\mathrm{m}|\mathrm{x})$ to get sample pairs $\{\mathrm{x}_i, \mathrm{z}_i, \tilde{\mathrm{m}}_i\}_{i=1}^n$
9:         Compute $L_{ELBO} = \frac{1}{n}\sum_{i=1}^n\left[\log p_g(\mathrm{x}_i, \mathrm{z}_i|\mathrm{m}_i) + \mathrm{KL}(q_\alpha(\mathrm{m}|\mathrm{x}_i)\|p(\mathrm{m}))\right]$ via Eq.13.
10:        Compute $L_G = L_G + L_{ELBO}$
11:     **end if**
12:    Update $G, q_\alpha$ (if using ERM) by minimizing $L_G$
    ▷ EBM training
13:    Apply data augmentation to generated samples: $\mathrm{x}_{\mathrm{aug}_i} = v(G(\mathrm{m}_i)), v \sim \mathcal{V}$ with probability $p$ and $G(\mathrm{m}_i)$ otherwise
14:    Sample negative examples $\{\tilde{\mathrm{x}}_i^T\}_{i=1}^n \sim p_\theta(\mathbf{x})$ initialized from $\tilde{\mathrm{x}}_i^0 = \mathrm{x}_{\mathrm{aug}_i}$ using Eq.9.
15:    Compute $L_{\mathrm{EBM}} \leftarrow \frac{1}{n}\sum_{i=1}^n E_\theta(\mathrm{x}_i, \mathrm{z}_i) - E_\theta(\tilde{\mathrm{x}}_i^T)$
16:    Update $E_\theta$ by minimizing $\mathcal{L}_{\mathrm{EBM}}$
17: **end for**

---

## A.6 INFERENCE MODEL FOR EM SETTING

We train an inference model for the EM setting using the following loss:

$$-\mathbb{E}_{p_{\mathrm{data}}(\mathrm{x},\mathrm{z})}\mathbb{E}_{q_\alpha(\mathrm{m}|\mathrm{x})}\left[\log p_g(\mathrm{z}|\mathrm{m}) - \frac{q_\alpha(\mathrm{m}|\mathrm{x})}{p(\mathrm{m})}\right], \tag{26}$$

where $p_g(\mathrm{z}|\mathrm{m}) = \rho\,\mathrm{sim}(\mathrm{z}, h(G(\mathrm{m}))$ by definition in Eq.13. The loss essentially minimizes the KL divergence between two conditional distributions: $\mathrm{KL}\left(p_{\mathrm{data}}(\mathrm{z}|\mathrm{x})\|p_g(\mathrm{z}|\mathrm{x})\right)$. This approach ensures feature preservation in the latent space rather than enforcing pixel-level reconstruction. Fig.9 compares reconstruction results using Eq.26 versus the traditional ELBO loss in VAEs. The traditional

ELBO fails to produce clear, semantically meaningful images with recognizable objects. Our training loss achieves high-quality reconstructions that preserve semantic properties of the input, such as object class, color, and visual style, without enforcing exact image reproduction. This indicates that our latent representation supports flexible instance generation.

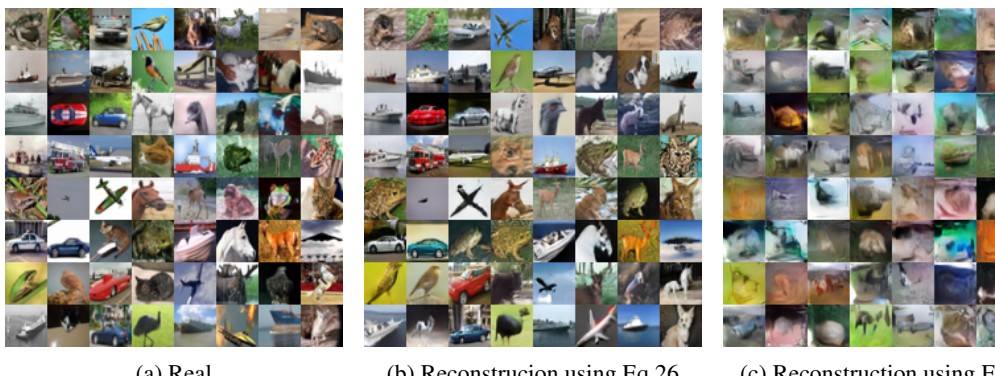

| (a) Real | (b) Reconstrucion using Eq.26 | (c) Reconstruction using ELBO |

Figure 9: Reconstruction with different training losses.

## A.7 RECONSTRUCTION OF LIEBM-ERM

While our autoencoder-style ERM scheme is designed primarily for initialization, we additionally demonstrate its image reconstruction capabilities in Figs.10 and 11. Following the test setting in Han et al. (2019), we also compare our approach with other models that also incorporate an inferential mechanism, where performance is quantitatively measured by per-pixel mean square error (MSE). As shown in Tab.11, our model achieves the best performance on CIFAR-10, outperforming Dual-MCMC even with Langevin refinement. On CelebA 64, our model achieves comparable results to Dual-MCMC but without requiring additional Langevin dynamics.

Table 11: Reconstruction evaluation using MSE on CIFAR-10 and CelebA 64. Inf+L=10 denotes using 10-step Langevin dynamics initialized by the inference model.

| Methods | CIFAR-10 | CelebA-64 |
|---|---|---|
| WS (Hinton et al., 1995) | 0.058 | 0.152 |
| VAE (Kingma & Welling, 2014) | 0.037 | 0.039 |
| ALI (Dumoulin et al., 2016) | 0.311 | 0.519 |
| ALICE (Li et al., 2017) | 0.034 | 0.046 |
| Divergence Triangle (Han et al., 2019) | 0.028 | 0.030 |
| Dual-MCMC (Inf) (Cui & Han, 2023) | 0.049 | 0.022 |
| Dual-MCMC (Inf+L=10) (Cui & Han, 2023) | 0.024 | **0.013** |
| **LIEBM-ERM** (Inf) | **0.019** | 0.014 |

## A.8 HYPERPARAMETER SETTINGS

We specify the hyperparameters used for our training on each dataset in Tab.12. We adopt two forms of function $F$ in Eq.5 for different datasets based on the generation performance. For CIFAR-10 and ImageNet 32, we define $F\left(f_\phi(\mathrm{x})\right) = \frac{-\|f_\phi(\mathrm{x})\|_2^2}{2}$, while for CelebA 64 and CelebA-HQ 256, we define $F$ to be a learnable linear function, which is trained along with $E_\theta(\mathrm{x}, z)$. The output dimension of $f_\phi(\mathrm{x})$ is 512.

## A.9 ADDITIONAL RESULTS

We provide more qualitative visual results for both EM and ERM settings in Figs.12-15.

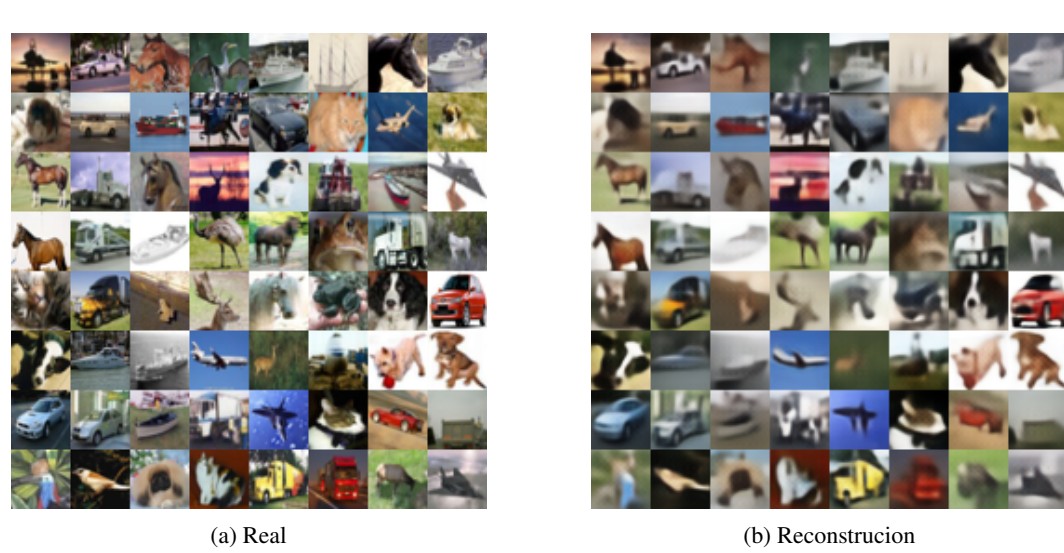

(a) Real                                    (b) Reconstrucion

Figure 10: Reconstruction of LIEBM-ERM on CIFAR-10.

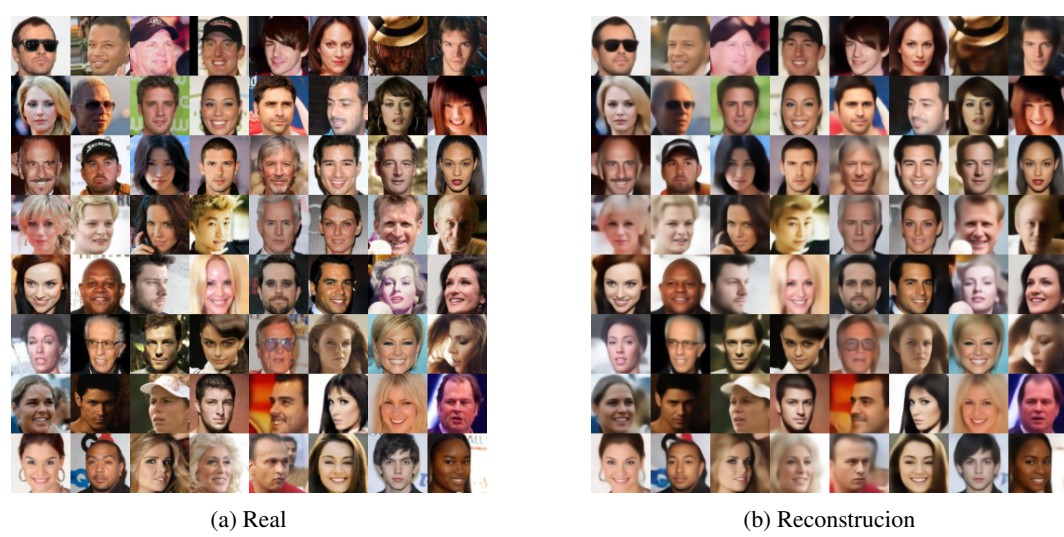

(a) Real                                    (b) Reconstrucion

Figure 11: Reconstruction of LIEBM-ERM on CelebA 64.

Table 12: Hyperparameters for each dataset.

|  | CIFAR-10 | ImageNet 32 | CelebA 64 | CelebA-HQ 256 |
|---|---|---|---|---|
| $E_\theta$ learning rate / Adam $\beta_1, \beta_2$ | 1e-4 / (0.0, 0.999) | 1e-4 / (0.0, 0.999) | 1e-4 / (0.0, 0.9) | 1e-4 / (0.0, 0.9) |
| $G$ learning rate / Adam $\beta_1, \beta_2$ | 2e-4 / (0.0, 0.9) | 2e-4 / (0.0, 0.9) | 3e-4 / (0.0, 0.9) | 3e-4 / (0.0, 0.9) |
| $q_\alpha$ learning rate / Adam $\beta_1, \beta_2$ | 2e-4 / (0.0, 0.9) | 2e-4 / (0.0, 0.9) | 1e-4 / (0.0, 0.9) | 1e-4 / (0.0, 0.9) |
| EMA decay rate | 0.9999 | 0.9999 | 0.9999 | 0.9999 |
| $\gamma$ for training | 0.01 | 0.01 | 0.01 | 0.01 |
| $\gamma$ for OOD | 0.1 | 0.1 | 0.1 | 1 |
| batch size | 256 | 256 | 256 | 128 |
| MCMC steps | 15 | 15 | 15 | 15 |
| MCMC step size $\delta^2$ | 25 | 25 | 0.1 | 0.1 |
| $\omega_1$ / $\omega_2$ in Eq.11 | 1 / 0.1 | 1 / 0.1 | 70 / 1 | 70 / 1 |
| $\rho$ in Eq.13 | 1 | 1 | 1 | 50 |
| training epochs | 500 | 100 | 300 | 300 |
| data range | [0, 1] | [-1, 1] | [-1, 1] | [-1, 1] |
| latent dimension | 128 | 128 | 128 | 256 |
| $E_\theta, G$ hidden channels | 256 | 512 | 1024 | 1024 |
| $q_\alpha$ hidden channels | 128 | 128 | 128 | 64 |
| $G$ params | 4.3M | 16.0M | 12.2M | 34.3M |
| $E_\theta$ params | 4.9M | 17.6M | 20.7M | 40.7M |
| $q_\alpha$ params | 15.2M | 15.2M | 15.2M | 8.1M |

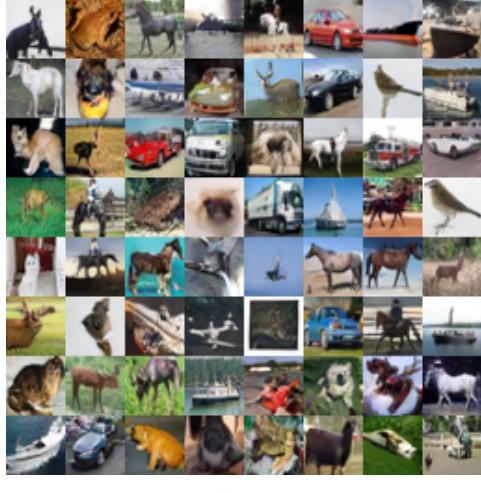
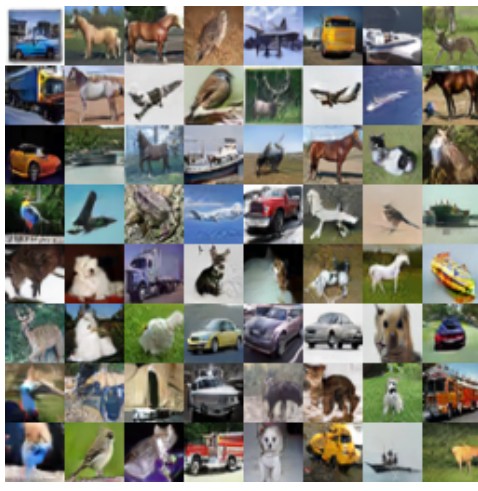

(a) EM    (b) ERM

Figure 12: Samples generated by LIEBM with MCMC refinement on CIFAR-10.

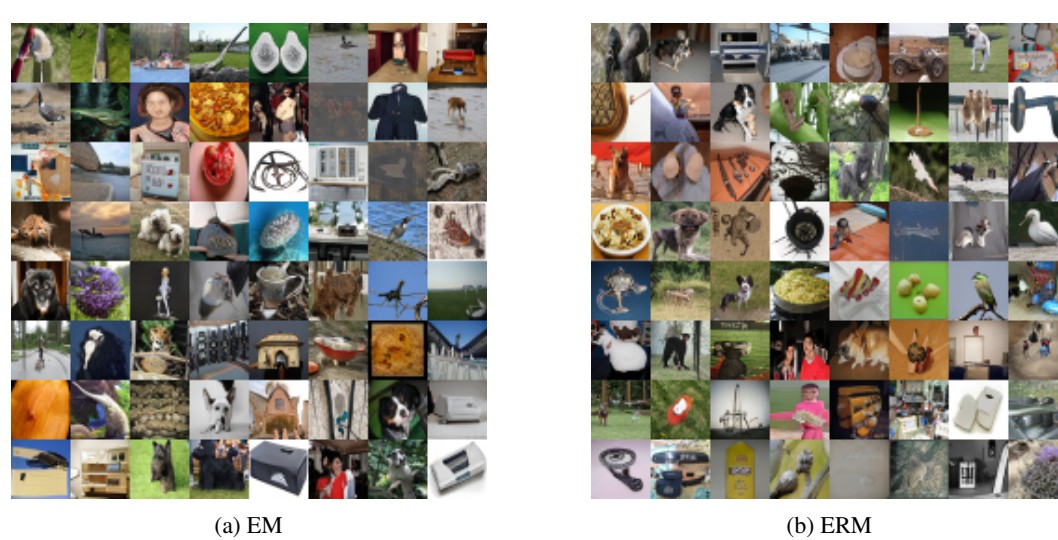

(a) EM        (b) ERM

Figure 13: Samples generated by LIEBM with MCMC refinement on ImageNet 32.

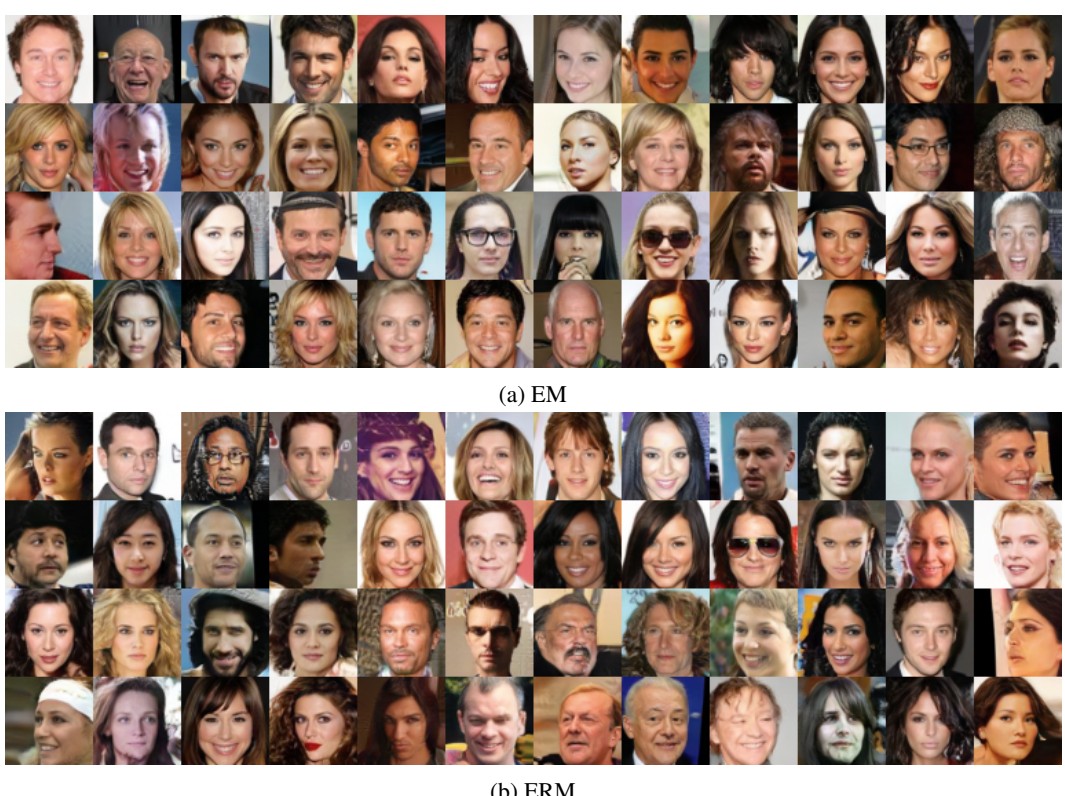

(a) EM

(b) ERM

Figure 14: Samples generated by LIEBM with MCMC refinement on CelebA 64.

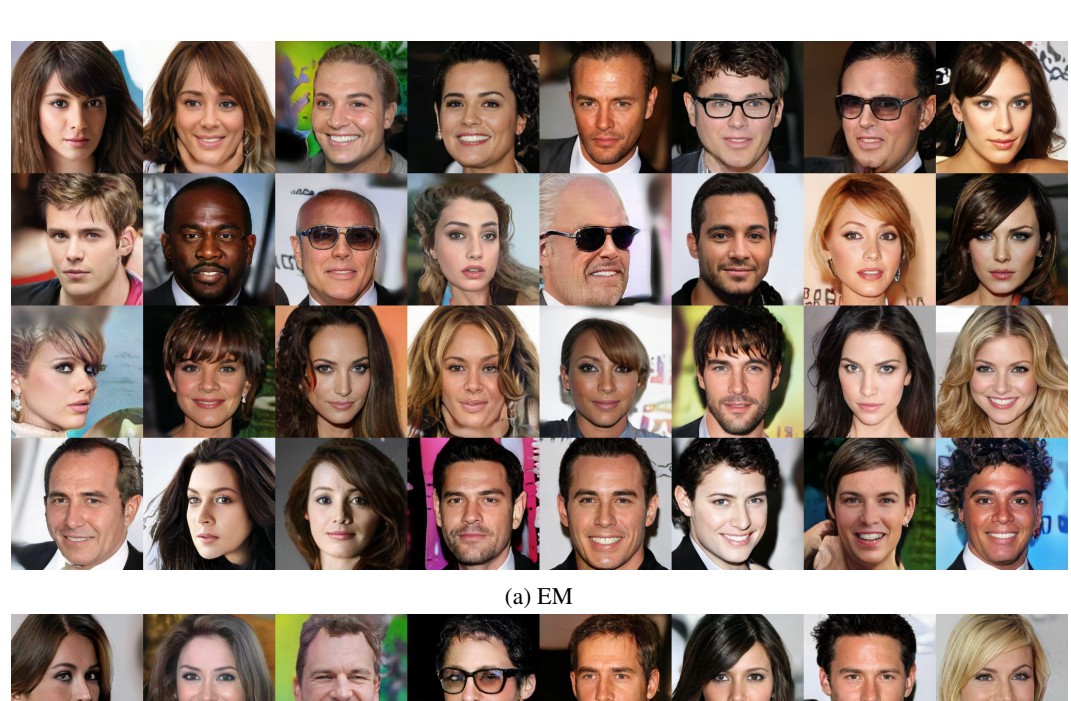

(a) EM

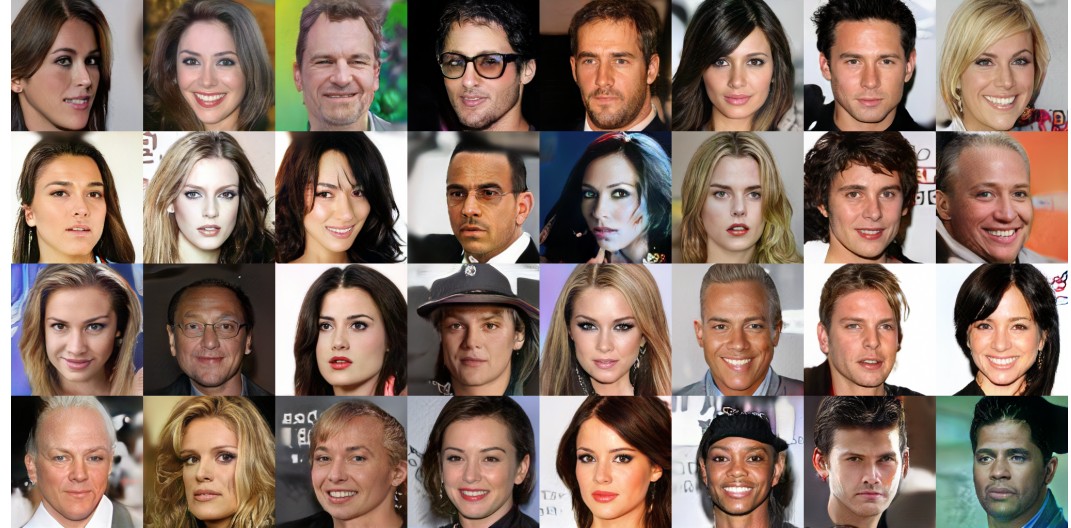

(b) ERM

Figure 15: Samples generated by LIEBM with MCMC refinement on CelebA-HQ 256.

