# OpenReview forum: "Latent-Informed Energy-Based Models with collaborative generator training"
_ICLR.cc/2026/Conference — Submitted to ICLR 2026_

### Official Review · Reviewer_xbAd · 2025-10-27

**Soundness:** 2
**Presentation:** 3
**Contribution:** 1
**Rating:** 2
**Confidence:** 4

**Summary:**

The paper proposes a Latent-Informed Energy-Based Model (LIEBM) that integrates a pretrained self-supervised encoder to a en EBM (as in CLEL (lee et al. 2023)) and uses cooperative training approach to improve training and sampling efficiency of EBMs (as in Xie et al. 2020).
Specifically, it formulates a latent-variable EBM where the latent codes are derived from a fixed self-supervised representation, enabling the energy function to capture both data distribution and semantic structure. The model is trained jointly with an auxiliary generator that initializes MCMC sampling through a single-step transformation, reducing the need for long-run chains. Two generator training strategies are explored — energy distribution matching (EM) and energy-real distribution matching (ERM) — designed to promote stable and collaborative optimization between the energy model and generator. The approach is empirically evaluated on several datasets for image generation, out-of-distribution detection, conditional sampling, and zero-shot image restoration, showing competitive quantitative performance compared to prior EBM variants.

**Strengths:**

The paper is well-motivated by a clear practical challenge in training energy-based models, namely the inefficiency of long-run MCMC sampling. It offers an interesting empirical exploration of combining pretrained self-supervised representations with cooperative training to stabilize and accelerate EBM learning.
The experimental section is extensive and covers diverse downstream tasks — including unconditional generation, OOD detection, and zero-shot restoration — which demonstrates the versatility of the proposed framework in practice.
The integration of a semantic latent space into the EBM objective is conceptually appealing and aligns with current trends in using pretrained representations to enhance generative modeling.
The paper is also well-organized, with several quantitative comparisons to illustrate the potential benefits of the approach despite its methodological overlap with prior work.

**Weaknesses:**

- **Novelty:** The idea barely uses cooperative training which already relies on a generator to accelerate MCMC sampling from training an EBM. The only difference here is that they augment the observed sample X with a "latent" variable Z, which in fact is simply an encoded version of X using an encoder and a random augmentation. Hence, if one thinks about the augmented variable \tilde{X} = (X,Z) as the new observation. there is really no difference with cooperative sampling of dual-MCMC already introduced in prior works. Additionally, the idea of using latent constructed using a pre-trained encoder + augmentation is exactly the one from CLEL (lee et al. 2023). This makes the methodological contribution very weak.

- **Clarity:** The description of the generator training in section 3.2 is not clear enough. It heavily relies on the prior works in  Xie et al. 2020 (section 3.2.1)  on cooperative learning and Cui & Han (2023) on dual-MCMC (section 3.2.2) without providing enough context for the reader not familiar with these approaches. Notably, it remains unclear how the conditional p_{\theta}(x|m) is defined and whether, in equation 9, the initial sample of the chain x_t^0 is obtained using x_i^0 = G(m_i) or using  another randomly sampled value for m.  While the original papers provide clear explanations, the present one is hard to understand without reading those papers first. Providing a precise algorithmic description could help.

- **Erratic choice of baselines from an experiment to another:** Dual-MCMC appears in the table 1,2, 3 and 5 but not in table 4 and 6. Same for CLEL, it appears in table 1, 4, 5 but not in 2, 3 and 6. These are the most important baselines, other baselines exhibit the same treatment, they seem to randomly appear in some tables but not in others without clear justification.

**Questions:**

- In equation 9 is x_t^0 initialize using x_i^0 = G(m_i) or using another randomly sampled m? How is the conditional p_{\theta}(x|m) defined?

- What are the ground truth images in table 5?
- Why some important baselines such as Dual-MCMC and CLEL appear in some tables and not others?

---

> ### Author Response · Authors · 2025-11-21
> **Response to Reviewer xbAd**
>
> **Q. Novelty**
>
> We clarify our methodological contribution in response to all reviewers. Our model is fundamentally different from Dual-MCMC and goes beyond a simple extension of CLEL.
>
> **Q. Clarity**
>
> We have added an algorithm description in Appendix A. 5 to facilitate better understanding of our method.
>
> [link: algorithm](https://ibb.co/JRkfQ08F)
>
> **Q. Erratic choice of baselines**
>
> Thanks for your careful review. Regarding baselines, then there is always a curation at play. As explained in our manuscript, CIFAR-10 is the primary benchmark for EBMs, while baselines on other datasets are less established. We evaluate our method on four common datasets, including CIFAR-10, ImageNet 32, CelebA 64, and CelebA-HQ 256, covering a broader range than most EBMs. For the tables where Dual-MCMC and CLEL are missing, these methods did not report experiments on those datasets and provide no network architectures or parameter settings. Therefore, we include their official results wherever available. Since our method is not simply an extension of Dual-MCMC or CLEL, we believe this comparison is reasonable.
>
> **Q. Ambiguity in Eq.9**
>
> $x\_i^t$ is initialized using $G(\mathrm{m}\_i)$. And $p_\theta\mathrm(x|m)$ follows a Gaussian distribution with fixed variance $\tau^2$. Our algorithm may help clarify the details.
>
> **Q. Ground truth images in Table 5**
>
> We assume you are referring to Fig.5 rather than Tab.5, as the meaning of ‘ground-truth images’ in Tab.5 is unclear. If you indeed meant Fig.5, we have now added the ground-truth images.
>
> **Moreover, as shown in our shared response to all reviewers, we have added more comparisons with other generative models. Our method gets SOTA among EBMs and further narrows the performance gap between EBMs and other generative models.**

---

> ### Author Response · Authors · 2025-11-26
> **Added CLEL baselines**
>
> We added comparisons with CLEL on additional datasets in our revised manuscript, even though these datasets were not included in the original paper, and no official network architecture or parameter settings were provided. Although our model is not simply an extension of CLEL, we do use a similar form of the energy function. For CelebAHQ 256, we trained CLEL using both their public architecture and our own, but the training did not converge. We also found that jointly training the energy function and the latent encoder requires significantly more memory than our approach (about 70 GB with batch size 32, compared to 40 GB with batch size 128 for ours). As for Dual-MCMC, its methodology is fundamentally different from ours (please see our response to all reviewers). It is also more time- and memory-consuming due to performing MCMC sampling in both the latent and data spaces and requiring twice as many sampling steps. Therefore, we only report its official results.

---

> > ### Comment · Reviewer_xbAd · 2025-11-26
> >
> > I thank the authors for their response.
> > I still think the main weakness is the lack of novelty, a point shared by all the other reviewers.
> > Although the authors explain in their response how their method is different, the differences seem rather minor. For instance, while the authors argue that joint energy function is a general formulation that recovers that of CLEL as a particular case when using cosine similarity, they also found that such choice was the top performing. In other words, it is unclear what concrete benefits does this level of generality brings.
> > I maintain my score.

---

> > > ### Author Response · Authors · 2025-12-03
> > > **Response to Reviewer xbAd**
> > >
> > > Thanks again for your feedback. The review discussion stage is coming to an end, and we are no longer able to engage in further discussion. We have added a new global response to better demonstrate the generality of our method. We suspect that the perceived lack of novelty may stem from our earlier wording, and we have revised the manuscript for clarity. We hope these improvements may change your view of our work.

---

> ### Author Response · Authors · 2025-11-27
> **Response to reviewer xbAd**
>
> We greatly appreciate your feedback. For our general formulation, we argue that it theoretically encompasses most existing latent-variable EBM variants, but we empirically find that the cosine-similarity form performs best under collaborative training and undermines the long-standing dominance of Gaussian latents. We also observe that Gaussian latents perform poorly on OOD data, suggesting limited flexibility in shaping the energy landscape.
>
> Moreover, the energy form is only one component of our method. Collaborative training with an auxiliary generator is equally crucial—something CLEL does not include. This collaborative scheme significantly improves both sampling speed and generation quality. In contrast, CLEL jointly trains the energy model and latent encoder, and its performance depends heavily on this joint optimization. We found this joint training to be far more memory-consuming than ours (see our second response to you). Our approach avoids such joint training while improving both generation and OOD performance. We are concerned that some of our contributions may have been overlooked, as the discussion centered primarily on the energy form.
>
> **We humbly request that the reviewer reconsider the rating.**

---

### Official Review · Reviewer_BDpZ · 2025-11-01

**Soundness:** 2
**Presentation:** 3
**Contribution:** 2
**Rating:** 4
**Confidence:** 4

**Summary:**

The paper tackles the issues such as mode collapse and exact alignment in generative prior with the utilization of pre-trained self-supervised representations as independent variables to guide the energy functions. The authors claim that the method achieves good quality across multiple tasks with lightweight architectures.

**Strengths:**

1. The paper is well written and easy to follow.
2. The paper addresses the prior-hole problem in latent EBMs with self-supervised pretrained spherical latent encoder, thus decoupling the semantic latent space from the generative prior..
3. The paper addresses that they achieved good quality performance in multiple tasks with lightweight architectures.

**Weaknesses:**

1. Limited novelty and incremental improvement: The proposed LIEBM addresses the  mode collapse and exact alignment in generative prior with the utilization of pre-trained self-supervised representations as independent variables. However, frameworks such as Energy Matching [1], VAPO [2] and Action Matching [3] eliminate the need for auxiliary models and avoid adversarial instability by formulating training as a variational PDE problem. These methods provide cleaner and general formulations, but not discussed and referenced in detail, which weakens the fundamental formulation of the proposed framework.
2. Incomplete ablation study: The authors claim that spherical latent posterior is better than gaussian posterior. But there’s no experimental evidence found to support this statement.
3. Validation of lightweight claims: There’s no results of computational metrics (cost/time/etc) that demonstrate the lightweight characteristics of the proposed model.
4. Incomplete comparison of EBMs baselines: The work does not provide a comprehensive comparison with other EBM-based models such as Energymatching and VAPO. Lack of engagement with these SOTA limits the impact of the proposed framework.

References:
[1] Balcerak, Michal, et al. "Energy Matching: Unifying Flow Matching and Energy-Based Models for Generative Modeling." NeurIPS 2025.

[2] Loo, Junn Yong, et al. "Learning Energy-Based Generative Models via Potential Flow: A Variational Principle Approach to Probability Density Homotopy Matching." TMLR 2025.

[3] Neklyudov, Kirill, Daniel Severo, and Alireza Makhzani. "Action Matching: A Variational Method for Learning Stochastic Dynamics from Samples." ICLR 2023.

**Questions:**

Please consider addressing the weaknesses noted above.

---

> ### Author Response · Authors · 2025-11-21
> **Response to Reviewer BDpZ**
>
> **Q. Limited novelty and incremental improvement**
>
> Thanks for your insightful comment! We have cited your mentioned paper in our manuscript and provided additional comparisons with these methods in our global response to all reviewers. For our fundamental formulation, we acknowledge that the papers you mentioned provide cleaner formulations inherited from diffusion or flow models, but they all require hundreds of sampling steps to match our performance. An auxiliary generator is known to accelerate sampling. Transforming those models into latent-informed versions with an auxiliary generator would be a promising direction! Thus, these methods and ours are complementary, not contradictory. Additionally, our model isn't an adversarial EBM and doesn't suffer from adversarial training instability.
>
> **Q. Superiority of our latent posterior**
>
> Thanks for your meticulous review! New experimental results supporting our latent posterior are included in our response to all reviewers.
>
> **Q. Validation of lightweight claims**
>
> We add both the inference time and parameter count below for comparison to validate the lightweight characteristics of our model.
> | Method | Time(s)&darr; | Params(M)&darr; | FID&darr; | &nbsp;GPU-Type |
> |--------|-------|-------|-------|------|
> | NVAE | 0.36 | &nbsp;- | 50.97 |&nbsp;&nbsp;V100 |
> | StyleGAN2 w/ ADA | 0.04 | 40 | 2.92 | &nbsp;&nbsp;V100 |
> | DDPM| 80.5 | 36 | 3.17 | &nbsp;&nbsp;V100 |
> | NCSN++ | 423.2 | 107 | 2.20 | &nbsp;&nbsp;V100 |
> | EBM-Based | | | |
> | VAEBM | 8.79 | 136 | 12.19 | &nbsp;&nbsp;V100 |
> | EC-VAE | 0.21 | 101 | 5.20 | &nbsp;&nbsp;RTX 2080 Ti |
> | CLEL | 82.05 | 31 | 8.61 | &nbsp;&nbsp;RTX 2080 Ti |
> | Dual MCMC | 9.32 | 48 | 9.26 | &nbsp;&nbsp;RTX 2080 Ti |
> | LIEBM w/o MCMC(ours) | 0.08 | 9 | 4.96 | &nbsp;&nbsp;RTX 2080 Ti |
> | LIEBM(ours) | 1.24 | 9 | 4.26 | &nbsp;&nbsp;RTX 2080 Ti |
>
> **Q.  Incomplete comparison of EBMs baselines**
>
> Thanks for your reminder! We provide additional comparisons, including your mentioned papers, in our response to all reviewers.

---

### Official Review · Reviewer_kJhT · 2025-11-04

**Soundness:** 3
**Presentation:** 3
**Contribution:** 2
**Rating:** 6
**Confidence:** 5

**Summary:**

This work introduces a method for learning a latent EBM. During sampling, a generator network provides an initialization of the sample, which is then further refined through MCMC sampling for a few steps. To learn such a model, ideas are adapted from recent EBM works CLEL, Cooperative Learning, and Dual MCMC. The joint distribution of observed and latent variables follows CLEL and obtains z|x using a random augmentation on x and encoding it to a z on the unit sphere using a pretrained contrastive model. Learning the latent energy function uses the CLEL closed-form distribution of z|x that measure the alignment between z and a learnable contrastive encoding of x, which is also used to define the energy function. A generator is used to initialize MCMC samples and trained in an alternating manner with the energy function using a reconstruction loss to push generator samples closer to the energy refinement. Finally, a inference network is introduced to also encourage the generator to match the distribution of the training data with a variational loss based on the Dual MCMC EBMs. Experiments show strong performance on CIFAR-10 and Celeb-A relative to other EBM methods.

**Strengths:**

* This work achieves strong generative and among EBMs.
* The method brings together several different directions in EBM learning to synthesize their strengths into a single model.
* The proposed model has good OOD performance relative to other EBM methods.

**Weaknesses:**

* There is not much methodological novelty in the paper. Learning the energy function uses the framework of CLEL, except that a pretrained encoder is used rather than a learned encoder. Cooperative learning has been used in the same way as this paper in a variety of previous works. The work is essentially a straightforward combination of CLEL and cooperative learning.
* The method requires a large number of interacting models: a pretrained SimCLR encoder, a learned encoder $f_\phi$ and $g_\psi$, a generator network $G$, and a separate inference network. This contrasts with the relatively simple setup of diffusion models.
* The empirical results still lag behind GAN and diffusion models. There is no investigation into whether the proposed methods can scale well with high dimensional and more complex distributions like high resolution ImageNet.
* The OOD comparisons might not be entirely fair, because the OOD calculation uses a network $h$ that is pretrained on large scale data as a contrastive model, rather than using only models learned from the generative process and target dataset as done in prior works.
* The distribution $p_{data} (z | x)$ in its current form is somewhat arbitrary/heuristic since it depends on a set of hand-crafted augmentations

**Questions:**

* Does the parameter count in Figure 2 include the energy function and generator only? I assume it does not include models that are unused for inference, like the pretrained encoder and variational network. How many parameters are used for the energy, and how many for the generator?
* Can you compare the inference time rather than parameter count for the methods listed?
* What function is used for $g$ in the formula $g(f_\phi (x))$?

---

> ### Author Response · Authors · 2025-11-21
> **Reviewer kJhT**
>
> **Q.  Lack of methodological novelty**
>
> We refer to our reply to all reviewers for a clear explanation of our contributions.
>
> **Q. A large number of interacting models**
>
> This is a point worth examining! There is a trade-off here. Although our LIEBM uses more networks than diffusion models, the auxiliary generator accelerates sampling, and the inference model supports low-dimensional feature extraction and reconstruction. In practice, if reconstruction is not required, our method trains only two networks; otherwise, it trains three. The size of our pretrained encoder is flexible (in our experiments is 12M), and using pretrained representation guidance is common in recent diffusion models such as REPA[1] and REG[2]. In future work, we may explore how to merge the inference model with the pretrained encoder.
>
> **Q. Lag behind GAN and diffusion models**
>
> To narrow the gap between LIEBM and other generative models, we update our architecture and achieve performance comparable to GANs and diffusion models. Please see our response to all reviewers for more results. For the ImageNet dataset, 32$\times$32 is the most commonly used resolution for EBMs, and baselines for ImageNet 256 are scarce. We evaluated our model on four commonly used datasets, covering a broader range than most EBMs, including Energy matching[3] and CDRL. There should be no theoretical issues with scaling to ImageNet 256. Given the substantial computational cost of ImageNet 256, we leave evaluation on this dataset to follow-up work.
>
> **Q. OOD comparisons**
>
> Our pretrained encoder is trained on the target dataset rather than large-scale data, so the comparisons are completely fair without data leakage.
>
> **Q. Arbitrary/heuristic $p\_{\text{data}}(\mathrm{z|x})$**
>
> Our definition of $p\_{\text{data}}(\mathrm{z|x})$ is heuristic rather than arbitrary. We use the same random augmentations as SSL methods, keeping the latent variables consistent with the SSL strategy of making augmented samples close in representation space. We also adopt another way to define $V$ for sampling from $p\_{\text{data}}(\mathrm{z|x})$: adding minor uniform noise via $\mathrm{x} = \frac{255}{256}\mathrm{x} + \mathrm{z}$. The performance is similar to the random augmentation method (See Sec.4.5 in our revised manuscript). This shows that our method is independent of the set of augmentations. More insightful design of $p\_{\text{data}}(\mathrm{z|x})$ can be explored in follow-up research.
>
> **Q. Parameter count**
>
> In Fig.2, EM counts the energy network and generator, while ERM adds the inference model on top. The individual parameter count for energy function and generator is reported in Tab.11 in Appendix (4.3M generator and 4.9M energy function).
>
> **Q. Inference time**
>
> We add both the inference time and parameter count below for comparison.
> | Method | Time(s)&darr; | Params(M)&darr; | FID&darr; | &nbsp;GPU-Type |
> |--------|-------|-------|-------|------|
> | NVAE | 0.36 | &nbsp;- | 50.97 |&nbsp;&nbsp;V100 |
> | StyleGAN2 w/ ADA | 0.04 | 40 | 2.92 | &nbsp;&nbsp;V100 |
> | DDPM| 80.5 | 36 | 3.17 | &nbsp;&nbsp;V100 |
> | NCSN++ | 423.2 | 107 | 2.20 | &nbsp;&nbsp;V100 |
> | EBM-Based | | | |
> | VAEBM | 8.79 | 136 | 12.19 | &nbsp;&nbsp;V100 |
> | EC-VAE | 0.21 | 101 | 5.20 | &nbsp;&nbsp;RTX 2080 Ti |
> | CLEL | 82.05 | 31 | 8.61 | &nbsp;&nbsp;RTX 2080 Ti |
> | Dual MCMC | 9.32 | 48 | 9.26 | &nbsp;&nbsp;RTX 2080 Ti |
> | LIEBM w/o MCMC(ours) | 0.08 | 9 | 4.96 | &nbsp;&nbsp;RTX 2080 Ti |
> | LIEBM(ours) | 1.24 | 9 | 4.26 | &nbsp;&nbsp;RTX 2080 Ti |
>
> **Q. Definition of $g$**
>
> The definition of $g$ is in Appendix A.8. Following the other reviewer's suggestion, we changed the symbol $g$ to $F$ in the revised manuscript to avoid confusion with $g_\psi$ in Eq.7.
>
> Ref: [1] Yu, Sihyun, et al. "Representation alignment for generation: Training diffusion transformers is easier than you think." ICLR 2025.
>
> [2] Wu, Ge, et al. "Representation Entanglement for Generation: Training Diffusion Transformers Is Much Easier Than You Think." NeurIPS 2025.
>
> [3] Balcerak, Michal, et al. "Energy Matching: Unifying Flow Matching and Energy-Based Models for Generative Modeling." NeurIPS 2025.

---

### Official Review · Reviewer_igVn · 2025-11-06

**Soundness:** 3
**Presentation:** 2
**Contribution:** 2
**Rating:** 4
**Confidence:** 4

**Summary:**

This paper proposes an energy-based model (EBM) framework that effectively trains a latent-variable EBM by leveraging (i) pretrained self-supervised models and (ii) cooperative learning with an auxiliary generator. The pretrained model helps the EBM efficiently capture semantic information, while the auxiliary generator accelerates MCMC sampling. Experimental results demonstrate the superiority of the proposed method over existing EBMs in unconditional image generation, and further highlight its versatility across various downstream tasks, including out-of-distribution detection, conditional sampling, and image restoration.

**Strengths:**

- This paper effectively integrates existing EBM frameworks, including CLEL, cooperative learning, and Dual-MCMC.
- The proposed framework shows competitive or superior performance compared to EBM baselines on several benchmarks, such as CIFAR, ImageNet-32, and CelebA.
- The paper provides various analyses and ablation studies that demonstrate the effectiveness of the proposed EBM.

**Weaknesses:**

- Although this integration itself can be viewed as a contribution, the lack of methodological innovation may weaken the novelty of the work. I found that LIEBM is essentially identical to CLEL except for the use of pre-trained SSL models; the negative sample augmentation strategy has already been proposed in EBM-CD; and the EM process follows cooperative learning. Beyond integrating these components, what additional contribution does this paper make?
- CDRL-large achieves 3.68 FID on CIFAR-10, which should be reported in Table 1. With this result, the proposed EBM outperforms other EBMs on CelebA-HQ-256 and ImageNet-32, but not on CIFAR-10 or CelebA-64. This also weakens the claimed contribution.
- Scalability should be further tested. The proposed method has been evaluated only on high-resolution but less diverse data (i.e., CelebA-HQ) and on diverse but low-resolution data (i.e., ImageNet-32). Is there any empirical evidence that this model can scale to both diverse and high-resolution datasets, such as ImageNet-256? This point is particularly important nowadays, as many generative models have successfully demonstrated such potential.
- Minor comments and questions
  - Comparisons of training efficiency (e.g., training time and memory consumption) should include not only other EBMs but also other classes of generative models such as diffusion models.
  - Is it possible to leverage marginal MCMC (Eq. 9) and joint MCMC (Eq. 10) simultaneously to achieve both faster training and the ability to learn multimodal distributions? For example, one can use joint MCMC first at the early stage, and then use marginal MCMC for improving diversity.
  - I am also curious about the training stability. Since MCMC sampling often causes instability in EBM training, did you ever observe any issues (e.g., NaN) while training LIEBM?
  - After training the generator $G$, it is directly usable for generation. How is the generation quality?
- Typos and editorial comments
  - Most EBM literature uses the "negative" unnormalized log-likelihood for energy functions. So I recommend using $p_\theta(x,z)\propto\exp(-E_\theta(x,z))$.
  - In L116-117, is $v$ a random augmentation? $v$ and $\mathcal{V}$ appear for the first time there, but no explanation is given.
  - In Eq. (5), $g$ is defined as a mapping from $f_\phi(x)$ to the energy value. What exactly is $g$? Like CLEL, is it the norm of $f_\phi(x)$? Furthermore, the same notation $g$ is reused in Eq. (7), which is confusing. I suggest using a different symbol for clarity.
  - In L180, should it be $x_i^0=G(m)$ or $x_i^0=G(m_i)$?
  - In Eq. (13), I think $G(m)$ should be replaced by $x$.
  - In Eq. (14), I think $\log$ is omitted.
  - The energy functin $E_\theta$ and the inference model $q_\alpha$ are parameterized by $\theta$ and $\alpha$, respectively, but $G$ is not. Why?
  - There is no overall figure or and algorithmic description of the proposed method. Also, it is unclear how the models (i.e., energy function $E_\theta$, generator $G$, and inference model $q_\alpha$) are exactly trained. For example, are they trained alternatively or jointly? Which architectures are used for the models $E$, $G$, and $q$? Since implementation details are crucial in EBM training, they should be clearly described.

**Questions:**

See the weaknesses part.

---

> ### Author Response · Authors · 2025-11-21
> **Response to Reviewer igVn**
>
> **Q. Lack of methodological innovation**
>
> We provide a clear explanation of our contributions in our replies to all reviewers.
>
> **Q. Not SOTA on CIFAR-10 and CelebA-64**
>
> Thanks for your reminder! As shown in our responses to all reviewers, using the modified architecture significantly improves our CIFAR-10 results, surpassing CDRL-large while using only 34M parameters and 16 sampling steps, compared with CDRL-large’s 177M parameters and 96 steps. On CelebA-64, our performance is nearly on par with the best EC-VAE(2.97/2.71) while using far fewer parameters(48M/105M). Across other datasets, we outperform existing EBMs by a large margin. We believe these results sufficiently demonstrate the effectiveness of our method.
>
> **Q. Test scalability**
>
> We have evaluated our model on CIFAR-10, ImageNet 32, CelebA 64, and CelebA-HQ 256, covering a broader range than most EBMs, such as Energy matching[1] and CDRL. There are few, if any, EBM baselines on ImageNet 256. We agree that scaling our model to higher-dimensional complex data is very interesting, but this comes with a huge increase in computational demand. There should be no theoretical issues with scaling to such data. We are willing to explore our model’s potential on this dataset in follow-up research.
>
> **Q. Training efficiency**
>
> We add memory usage and training iterations below for comparison.
> | Method | Memory usage(GB)&darr; | training iterations(K)&darr; | FID&darr; |
> |--------|-------|-------|-------|
> | R3GAN[2] | 55 | 250 | 3.03 |
> | NCSN++ | 21 | 1300 | 2.38 |
> | EBM-Based | | | |
> | VAEBM | 129 | 25 | 12.2 |
> | CLEL | 10 | 100 | 8.61 |
> | CDRL | 69 | 400 | 4.31 |
> | LIEBM-EM(ours) | 14 | 100 | 4.26 |
>
> **Q.  Mixing training of EM and EJM**
>
> Thanks for your inspiring suggestion! We tried your proposal, but its performance was worse than training EM and EJM separately ($\approx$ 5). We suspect that EM and EJM rely on different intrinsic mechanisms that cannot be simply combined, further supporting the independence of the two settings.
>
> **Q. Training stability**
>
> This is a good question! Training instability has long been a major challenge for EBMs. Early on, we also frequently encountered NaN issues, such as those caused by Gaussian posteriors or step-size adaptation. With accumulated experience, our current approach rarely encounters NaNs. The modified version has also been tested with multiple energy-function networks and remains consistently stable.
>
> **Q. Generation quality for generator**
>
> We reported the generation results without energy-based refinement in Tab.1 of our manuscript. Even without refinement, our model outperforms most EBMs and even surpasses the 1-step consistency model and its variant ACT on CIFAR-10.
>
> **Typos and editorial comments**
>
> We have modified our manuscript accordingly. The architectures used for the models are specified in the first part of the Experiments section. Moreover, we have added an algorithmic description in Appdix A.5.
>
> [link: algorithm](https://ibb.co/JRkfQ08F)
>
> Ref: [1] Balcerak, Michal, et al. "Energy Matching: Unifying Flow Matching and Energy-Based Models for Generative Modeling." NeurIPS 2025.
>
> [2] Yiwen Huang, et al. “The GAN Is Dead; Long Live the GAN! A Modern Baseline GAN.” NeurIPS 2024.

---

### Author Response · Authors · 2025-11-21
**Global response to all reviewers**

We are grateful to all the reviewers for their valuable feedback, helpful suggestions, and insightful questions. We appreciate the recognition from reviewers regarding our well-motivated framework (**xbAd, BDpZ**) and comprehensive experimental validation (**igVn, kJhT, BDpZ, xbAd**). Here we address the common concern of limited novelty. In fact, we suspect that the perceived lack of novelty may stem from our earlier wording; our methodological novelty goes far beyond simply integrating existing EBM frameworks, such as CLEL, cooperative learning, and Dual-MCMC. We clarify our contributions below and will revise the manuscript accordingly.

---

> ### Author Response · Authors · 2025-11-21
> **Novelty Clarification**
>
> We clarify our contributions as follows:
> 1. Our main contribution is introducing a unified and efficient latent-variable EBM training paradigm and demonstrating the necessity of this joint space optimization. We theoretically derive that this paradigm only requires MCMC sampling in data space, avoiding expensive cost in augmented ($\mathrm{x,z}$) space.  Via Eq. 4, our joint energy function learns semantic data-latent relationships directly from real data, in a more reasonable way than previous contrastive learning between real and fake spaces in EBMs and GANs. We believe latent-variable models are crucial for EBM training, and our framework serves as an important baseline for future EBM research.
> 2. Our definition of joint energy function is a general formulation, not merely borrowed from CLEL. When the explicit posterior is defined as Gaussian, our $E_\theta(\mathrm{x,z})$ reduces to conventional formulations [1,2,3]. With cosine similarity form, it's similar to CLEL; With an exponential family form of $E\_{\theta, \lambda}(\mathrm{x, z})=\left\langle\lambda-t\_\theta(\mathrm{x}), \eta(\mathrm{z})\right\rangle+B(\lambda)$, it reduces to CEBM[4]. Thus, our definition in Eq.5-6 encompasses most existing latent-variable EBMs as special cases. We empirically compared these choices within our framework and found that only cosine similarity is stable and performs significantly better than others (See FID training curves below). Therefore, we adopt this form for our joint energy function. More expressive energy forms can be explored within our framework in future work.
>
> [link: FID training curves for different forms](https://ibb.co/1G59zdsV)
>
> 3. Our collaborative learning extends cooperative learning. While EM matches cooperative learning, our EJM and ERM settings are tailored for joint energy training. Notably, ERM differs from Dual-MCMC by redefining $p_g(\mathrm{x,z})$ through a prior variable $m$, where $p_g(\mathrm{x,z}|\mathrm{m})$ is a novel formulation. We provide a thorough comparison of these three settings and analyse their respective pros and cons.
> 4. Our pretrained encoder is different from that in CLEL, which is jointly trained with EBM. We found that this joint training may work well in CLEL, but impair our collaborative framework. Our design is theoretically more rigorous than CLEL and shares the same philosophy as recent works in diffusion models such as REPA[5] and REG[6].
> 5. Our negative sample augmentation strategy serves a fundamentally different purpose than EBM-CD. While EBM-CD augments all initial samples from the replay buffer to prevent sampling collapse. We apply augmentation to generated samples with a small probability $p$ to mitigate catastrophic forgetting in distant regions. We design this for a better energy landscape rather than sampling quality. We tested replay buffers[7,8], GAN augmentation[9], and real samples initialization[10], but only our augmentation strategy proved effective for collaborative training.
>
> **Practical training challenges have long hindered the mainstream adoption of EBMs in generative modeling. Our method offers a simple yet effective way to enhance model performance and stability, which we believe could be valuable to the community at this stage.**
>
> [1] Cui, Jiali, and Tian Han. "Learning energy-based model via dual-mcmc teaching." NeurIPS 2023.
>
> [2] Han, Tian, et al. "Joint training of variational auto-encoder and latent energy-based model." CVPR 2020.
>
> [3] Kan, Ge, et al. "Bi-level doubly variational learning for energy-based latent variable models."CVPR 2022.
>
> [4] Wu, Hao, et al. "Conjugate energy-based models." ICML 2021.
>
> [5] Yu, Sihyun, et al. "Representation alignment for generation: Training diffusion transformers is easier than you think." ICLR 2025.
>
> [6] Wu, Ge, et al. "Representation Entanglement for Generation: Training Diffusion Transformers Is Much Easier Than You Think." NeurIPS 2025.
>
> [7] Hill, M., et al. “Learning probabilistic models from generator latent spaces with Hat EBM.” NeurIPS 2022.
>
> [8] Yilun Du, et al. “Improved Contrastive Divergence Training of Energy-Based Model” ICML 2021.
>
> [9] Karras, Tero, et al. “Training Generative Adversarial Networks with Limited Data.” NeurIPS 2020.
>
> [10] Geoffrey E. Hinton, “Training Products of Experts by Minimizing Contrastive Divergence.” Neural Computation 2002.

---

> ### Author Response · Authors · 2025-11-21
> **More comparisons with other generative models**
>
> To bridge the performance gap with other generative models, we adopt advanced networks and meticulous parameter tuning to improve generation quality. As shown in the table below, with our modified architecture, our method achieves 3.28 on CIFAR-10, which gets SOTA among all EBMs and is comparable to GANs and diffusion models. Moreover, when substituting our energy function with those from CDRL or EC-VAE, our approach consistently surpasses them while using a more lightweight architecture, further demonstrating the robustness and superiority of our training framework. We believe our method has strong potential to drive further progress and spark more research in this area.
> | Method | FID&darr; | Params(M)&darr; | NFE&darr; |
> |--------|-------|-------|-------|
> | R3GAN[4] | 3.03 | 41 | 1 |
> | DDPM | 3.17 | 36 | 1000 |
> | **EBM-based** |
> | Action Matching[1] | 10.04 | &nbsp;- | 1000 |
> | VAPO[2] | 6.72 | 56 | 74 |
> | Energy Matching[3] | 3.34 | 50 | 325 |
> | EC-VAE | 5.20 | 101 | 1 |
> | CDRL| 4.31 | 73 | 96 |
> | CDRL-Large| 3.68 | 177 | 96 |
> | Ours(EC-VAE) | 4.27 | 46| 11 |
> | Ours(CDRL)| 3.86 | 56 | 16 |
> | Ours-modify | 3.28 | 34 | 16 |
>
> [1] Neklyudov, Kirill, Daniel Severo, and Alireza Makhzani. "Action Matching: A Variational Method for Learning Stochastic Dynamics from Samples." ICML 2023.
>
> [2] Loo, Junn Yong, et al. "Learning Energy-Based Generative Models via Potential Flow: A Variational Principle Approach to Probability Density Homotopy Matching." TMLR 2025.
>
> [3] Balcerak, Michal, et al. "Energy Matching: Unifying Flow Matching and Energy-Based Models for Generative Modeling." NeurIPS 2025.
>
> [4] Yiwen Huang, et al. “The GAN Is Dead; Long Live the GAN! A Modern Baseline GAN.” NeurIPS 2024.

---

### Author Response · Authors · 2025-11-26
**Any additional concerns?**

We appreciate the insightful suggestions from all reviewers. In response, we have conducted additional experiments and provided further clarifications of our contributions to comprehensively address the raised concerns. If any reviewer has further questions or concerns, we will try our best to address them.

---

### Author Response · Authors · 2025-12-03
**Global response to all**

We appreciate the constructive comments from all reviewers and the feedback from xbAd. As the review discussion stage is drawing to a close, we highlight what we believe are the reviewers’ primary concerns. We have updated our manuscript according to all reviews, as the misperception of limited novelty was likely due to our previous phrasing.
* For the generality of our method, we compare performance with different energy forms in the previous global response. At that time, we relaxed the latent variables to be unnormalized for Gaussian posterior and CEBM, which may be theoretically reasonable, but is practically unstable. We also try normalized latent variables, which align with our definition of $p\_{\text{data}}(\mathrm{z|x})$, and observe more stable performance. Due to the sensitive variance effects, we constrain the log-variance to $[-1,1]$ for Gaussian posterior and fix the Gaussian variance for CEBM, We observe that both performances are better than Gaussian-posterior Dual MCMC but worse than cosine-similarity form. If we further fix the variance in Gaussian posterior, its performance is very close to cosine-similarity form, it's not surprising since
$\left\|\mathrm{z_1}-\mathrm{z_2}\right\|_2^2=2-2 \operatorname{sim}(\mathrm{z_1},\mathrm{z_2})$. Therefore, we demonstrate that decoupling the implicit data energy from the explicit posterior, as in Eq.5, offers greater flexibility and easier control, and cosine-similarity form offers an alternative to the long-standing use of Gaussian latents. Overall, we reiterate that our model is a general formulation, not merely an extension of CLEL or other approaches.
* We also adopt another way to define $\mathcal{V}$ for sampling from $p\_{\text{data}}(\mathrm{z|x})$: adding minor uniform noise via $\mathrm{x} = \frac{255}{256}\mathrm{x} + \mathrm{z}$, where $z\sim U(0,\frac{1}{256})$. The performance is very close to the random augmentation method. This shows that our method is agnostic to the design of $\mathcal{V}$, which responds to the concerns for reviewer **kJhT**.

[link: FID curve with different $\mathcal{V}$ choices and energy forms](https://ibb.co/HD8G3dQv)

As a parting remark, most current generative models concentrate on data distribution modeling while largely neglecting geometric structure, which is also essential for understanding the data manifold. Our latent-informed EBM injects geometric information into the latent variables and enables the energy function to capture both the semantic geometry of the data manifold and the data likelihood. Modeling data distributions while understanding their geometric structure is both the foundational motivation and the ultimate objective of generative models. Moreover, as shown in the last global response to all reviewers, our latent-informed EBM (LIEBM) significantly improves EBM performance (achieves SOTA among all EBMs on CIFAR-10 (**FID 3.28 with 34M params in total**), ImageNet 32, and CelebAHQ 256 with lightweight architectures) and greatly narrows the gap with GANs and diffusion models. We believe our LIEBM training paradigm is essential for advancing EBM training and the broader generative modeling community.

---

### Meta-Review · Area_Chair_1sbx · 2026-01-06

**Summary:**

Reviewer igVn views the paper as technically solid and well-executed, but insufficiently novel, incomplete in empirical validation, and under-documented in key methodological aspects. These issues collectively justify a borderline rejection, with encouragement to strengthen novelty claims, expand scalability experiments, and improve clarity in a future revision.

Reviewer kJhT’s recommendation is tempered by several concerns. The primary issue is limited methodological novelty, as the approach largely constitutes a straightforward integration of CLEL-style energy learning and cooperative learning, with the use of a pretrained encoder viewed as an incremental modification rather than a conceptual advance. The method also introduces a relatively complex system with multiple interacting components, raising questions about practicality and inference efficiency compared to simpler paradigms such as diffusion models. Empirically, although competitive within the EBM family, the results still lag behind GANs and diffusion models, and the paper does not provide evidence of scalability to higher-resolution or more complex datasets. In addition, the fairness of the OOD evaluation is questioned due to reliance on a contrastive model pretrained on large-scale external data, and some modeling choices are perceived as heuristic.

Reviewer BDpZ: The paper is judged marginally below the acceptance threshold due to several substantive concerns. The primary issue is limited methodological novelty, as the proposed LIEBM framework addresses known challenges in latent EBMs using pretrained self-supervised representations, but does not sufficiently position or compare itself against closely related and more general EBM formulations such as Energy Matching and VAPO. In addition, key claims are insufficiently validated, including the asserted advantage of a spherical latent posterior over a Gaussian one, which is not supported by targeted ablation studies. The paper also lacks quantitative evidence for its “lightweight” characterization, as no computational cost or efficiency metrics are reported. Finally, the comparison with existing EBM baselines is incomplete, limiting the ability to assess the proposed method’s relative strengths within the broader EBM literature.

Reviewer xbAd: The reviewer concludes that the paper does not meet the bar for acceptance due to incremental contributions, insufficient methodological clarity, and inconsistent experimental comparisons.

**Reviewer Concerns:**

Reviewer igVn: The rebuttal successfully resolves many technical, empirical, and clarity-related concerns, notably regarding benchmark competitiveness, efficiency, stability, and implementation details. However, two core issues remain: (1) the limited degree of methodological novelty beyond integration, and (2) the lack of convincing empirical evidence for scalability to large, diverse, high-resolution datasets. These remaining concerns continue to weigh against a clear acceptance, keeping the paper in a borderline position despite a substantially improved rebuttal.

Reviewer kJhT: The rebuttal satisfactorily addresses several of the reviewer’s concerns: it clarifies the parameter breakdown and inference-time costs, demonstrating that LIEBM is relatively lightweight and efficient at inference compared to prior EBMs; it corrects a misunderstanding about the OOD setup by confirming that the pretrained encoder is trained only on the target dataset, alleviating fairness and data-leakage concerns; and it clarifies architectural details. The rebuttal also partially mitigates concerns about empirical performance by reporting improved results that narrow the gap with GANs and diffusion models and by adding explicit inference-time comparisons. However, some concerns remain outstanding: in particular, the rebuttal does not fundamentally resolve the reviewer’s core skepticism about limited methodological novelty, as the contribution is still largely framed as an integration of existing ideas rather than a new learning principle, and scalability to high-resolution, complex datasets (e.g., ImageNet-256) is argued theoretically but not empirically demonstrated. As a result, while clarity and empirical positioning have improved, questions about conceptual novelty and large-scale applicability persist.

Reviewer BDpZ: The rebuttal partially addresses the reviewer’s concerns. The authors provide a clearer positioning of their method relative to Energy Matching and VAPO, arguing complementarity rather than contradiction, and add new comparisons and citations, which helps mitigate the concern about limited novelty, as the core formulation remains incremental relative to prior EBM frameworks. The rebuttal adequately addresses the lightweight claim, as concrete inference-time and parameter-count comparisons are now provided and clearly demonstrate improved efficiency relative to several baselines. The concern regarding the superiority of the spherical latent posterior is partially addressed through the inclusion of new experimental results. Finally, the concern about incomplete EBM baseline comparisons is partially addressed by the provided additional comparisons.

Reviewer xbAd: The rebuttal successfully addresses several clarity-related and factual concerns, and it improves the transparency of the experimental setup. However, the central issue, such as limited novelty and insufficient differentiation from prior work, remains largely unresolved.

**Reviewer Scores:**

Reviewer igVn: Based on the substance of the rebuttal and the reviewer’s original concerns, the most likely outcome is a modest upward revision, but not a decisive change to a clear accept.

Given the rebuttal, Reviewer kJhT would likely maintain the original score of 6 (marginally above the acceptance threshold).

Given the rebuttal, Reviewer BDpZ would likely increase the score slightly, but not enough to cross the acceptance threshold.

Reviewer xbAd would likely maintain a negative recommendation, as the rebuttal does not fundamentally change the assessment of the paper’s contribution relative to existing literature.

All reviewers have a common concern regarding the incremental contribution. The AC think that the rebuttal clarifies the authors’ intended novelty and significantly improves the positioning of the work relative to CLEL, Dual-MCMC, and cooperative learning. While the framework offers a useful unification and strong empirical validation, the core conceptual advance remains subtle, and some reviewers may still view the contribution as incremental. As such, the novelty concerns are partially but not fully resolved. Therefore, the AC encourages the authors to resubmit this work to a future venue after carefully addressing the reviewers’ comments and incorporating the suggested revisions. While the paper is well motivated and shows promising empirical results, the current version does not yet meet the bar for acceptance. The AC recommends rejection at this time.

---

### Decision · Program_Chairs · 2026-01-26

Reject